# Land Cover Maps Production with High Resolution Satellite Image Time Series and Convolutional Neural Networks: Adaptations and Limits for Operational Systems

**Andrei Stoian [1], Vincent Poulain [2], Jordi Inglada [3,4,\*][ID], Victor Poughon [4][ID] and Dawa Derksen [3]**

[1]   Thales/SIX/ThereSiS, 91477 Palaiseau, France
[2]   Thales Services, 31555 Toulouse, France
[3]   Centre d'Etudes Spatiales de la BIOsphere (CESBIO), Université de Toulouse,
      CNES/CNRS/IRD/UPS/INRA, 31555 Toulouse, France
[4]   Centre National d'Etudes Spatiales (CNES), 31555 Toulouse, France
**\***   Correspondence: jordi.inglada@cesbio.eu

**Abstract:** The Sentinel-2 satellite mission offers high resolution multispectral time-series image data, enabling the production of detailed land cover maps globally. When mapping large territories, the trade-off between processing time and result quality is a central design decision. Currently, this machine learning task is usually performed using pixel-wise classification methods. However, the radical shift of the computer vision field away from hand-engineered image features and towards more automation by representation learning comes with many promises, including higher quality results and less engineering effort. In particular, convolutional neural networks learn features which take into account the context of the pixels and, therefore, a better representation of the data can be obtained. In this paper, we assess fully convolutional neural network architectures as replacements for a Random Forest classifier in an operational context for the production of high resolution land cover maps with Sentinel-2 time-series at the country scale. Our contributions include a framework for working with Sentinel-2 L2A time-series image data, an adaptation of the U-Net model (a fully convolutional neural network) for dealing with sparse annotation data while maintaining high resolution output, and an analysis of those results in the context of operational production of land cover maps. We conclude that fully convolutional neural networks can yield improved results with respect to pixel-wise Random Forest classifiers for classes where texture and context are pertinent. However, this new approach shows higher variability in quality across different landscapes and comes with a computational cost which could be to high for operational systems.

**Keywords:** land cover mapping; convolutional neural networks; UNET; Sentinel-2

## 1. Introduction

### 1.1. The Need for Land Cover Mapping

Land cover information is a crucial asset for many scientific and operational applications, such as vegetation monitoring, urban management, etc. Many of these applications need high resolution up to date land cover information. Land cover databases are freely available over wide areas, for example, Corine Land Cover (CLC) over the European Union [1] or ESA's CCI land cover products globally available. However, they often fail to meet the needs of many users: CLC has a minimum mapping unit of 25 ha and is only updated every 6 years, and ESA's Globcover cover maps have a resolution of 300 m [2].

Until recently, automatic land cover mapping using remote sensing imagery was only performed at coarse resolution (CCI, for instance). Higher resolution imagery was only used through photo-interpretation (as for CLC) leading to high costs and long production delays.

With the availability of Landsat and more recently Sentinel [3] image time-series, the possibility of accurate automatic land cover mapping at high resolution has become a reality [4]. Some one-shot initiatives have been published, as for instance the GLC-30 [5]. To the best of our knowledge, the only recurrent operational (large-scale, recurrent production) automatic high resolution land cover mapping is the French Theia Land Data Centre which delivers a 10 m resolution national land cover map with 17 classes and an accuracy of 89% [6,7]. The approach has to be fully automatic in order to allow the delivery of a map every year less than 3 months after the reference period.

The approach used by Theia is described in [8] and uses pixel-based supervised classification, leveraging existing (and therefore, out of date) data bases as annotations for training and all available Sentinel-2 imagery acquired during the mapping period.

This approach allows to deal with the following crucial aspects in order to achieve efficient and robust large-scale land cover mapping:

- automation for efficiency and timeliness;
- spatial continuity of the maps;
- temporal coherence between updates of the product;
- reproducibility of the results;
- support of changes of nomenclature without changing the system.

Although the approach presented in [8] is used for the production of annual land cover maps of metropolitan France, the recognition of some classes of the nomenclature, mainly those where pixel context is important, is not satisfactory (accuracies below 60%) and new classification approaches need to be explored.

## 1.2. The State-of-the-Art is Supervised Classification

The literature on land cover mapping has established supervised classification methods as the most efficient way of performing the task. Most papers in the literature work on small areas, particular sets of classes (non exhaustive nomenclatures) and perfect reference data for training. In this paper, we concentrate on operational approaches, that is the development and delivery of reliable and recurrent data products within a pre-defined time schedule [5]. This is different from experimental approaches for large area land cover mapping and monitoring [9,10], which focus on the development and performance testing of novel algorithms and models.

One of the main challenges for scaling land cover mapping to very large areas is dealing with the following issues:

1.  Reference data are costly to obtain and one has to rely on existing data bases which contain errors or are out of date.
2.  Reference data are sparse, only a small amount of the pixels are annotated.
3.  Large areas may present climatic and topographic variabilities introducing different behaviours for the same class across different landscapes.

## 1.3. The Promise of Deep Learning

Although the overall accuracy achieved with the methodology of [8] is very high and most of the classes are very well recognized, some important confusions remain, mostly for classes where the pixel context may be important. Discontinuous urban fabric, for instance, is a mix of buildings and vegetation, and therefore, pixel-based classification at 10 m resolution will have tendency to confuse it with continuous urban fabric or some of the vegetation classes. Other classes are very difficult to distinguish from each other because of their similarity (in terms of spectral and temporal signatures) or because of the inherent intra-class variability.

These limits of the current machine learning approach used (pixel-based, engineered features) may be overcome by the use of more powerful approaches recently made available. The recent developments in the field of deep learning (DL) and more precisely convolutional neural networks (CNN) need to be assessed as candidates for the improvement of fully automatic land-cover mapping using high resolution image time-series as those acquired by Sentinel-2.

The complexity and intra-class variability in the problem could be handled by the extraction of features of increasing semantic level in the DL structure. The spatial context of pixels could be taken into account by the convolutional structure of the networks.

For a comprehensive review of DL applied to Remote Sensing, we refer the reader to [11]. Although new approaches have been published since this review, the state-of-the-art has mainly evolved to more complex methods which are, therefore, need a large amount of ressources. Furthermore, most of the methods in the litterature are used on academic datasets (small areas, densely annotated) which are not representative of operational applications.

*1.4. Semantic Segmentation vs. Patch Based Methods*

Two kinds of CNN can be used for land cover mapping. Patch-based approaches assign a class to the center pixel of an image patch and are a straightforward application of image classification networks to the task of semantic segmentation (i.e., the classification of every pixel in an image). Semantic segmentation can also be addressed by specific network architectures (Segnet [12], U-Net [13]) which are more computationally efficient and can also learn spatial dependencies between classes. Segmentation networks, especially U-Net, have another advantage over patch methods, since the latter have discontinuities in their prediction which appear as label noise in the maps [14] and thus require a smoothing step in post-processing [15]. Fully convolutional networks like U-Net implicitly perform a regularization step similar to what Conditional Random Fields (CRF) do. However, a post-processing step using CRF increases complexity and is rather sensitive to hyper-parameter selection [16].

Volpi et al. [17] compare a simple fully convolutional patch based method to a downsample-upsample network similar to U-Net but without skip connections. They obtain state-of-the-art results on 10 cm resolution imagery with a densely annotated dataset.

In the case of operational land cover mapping of large geographical areas, compared with semantic segmentation techniques patch based approaches have an important disadvantage because of their lower computational efficiency at prediction time, since every pixel of the patch has to be classified.

Segmentation network architectures were initially applied to dense annotations (see [12] and references therein), in other words, all the pixels of the training image patches need to be annotated. In the case of sparse annotations (reference labels are given as polygons which may not be adjacent to each other, may represent only parts of the objects and cover a small portion of the area of interest), for patch based methods one simply needs to exclude all patches centered on non-annotated pixels. However, segmentation networks need to be adapted in order to exploit the local class correlations available in the sparse polygons [18].

One of the main drawbacks of segmentation CNNs, such as Segnet and U-Net, is their lack of geometric accuracy, i.e., the prediction masks are too smooth and do not show sharp straight borders between classes. This is most problematic in the case of cartographic products where classes have sharp edges (e.g., buildings, roads, field boundaries). To improve upon this situation, in [19], the authors introduced additional supervision in the form of a class edge detector into the final loss function. In [20], a modified U-Net with residual connections and a guided filter were used in post-processing to straighten building prediction mask edges. Simpler methods have also been proposed, where CNN predictions are refined with object-based image analysis approaches [21]. Although these approaches improve the geometric accuracy of segmentation CNNs, they rely on the availability of dense annotations in order to learn how to predict edges or to regularize the segmentation.

Practical large-scale reference data for land cover prediction, outside of academic datasets, are hard to obtain. Maggiori et al. [22] run an initial training pass on roughly annotated data automatically

generated from OpenStreetMap and then fine-tune the network on precise manual annotations. In this fashion, they train a multi-scale patch classifier on 10 3000 × 3000 pixels from 0.5 m resolution Pleiades images. Although this approach can be applied at the level of an individual scene, it does not scale to the country level because of the landscape and climatic differences.

Methods for multi-temporal multi-source land cover classification were proposed by Kussul et al. [23], learning a classifier for 7 × 7 patches with 54 channels, containing 19 temporal samples sourced from Landsat 8 and Sentinel 1A data. The approach was a straightforward stacking in the time dimension and a patch-based classification assigning the predicted label to the center pixel of the patch. No particular adaptation of the architecture for the temporal aspect of the data was proposed. Ji et al. [24] adapted the patch methods to better take into account the temporal dynamics by using 3D 3 × 3 × 3 convolutions (2 spatial dimensions and 1 time dimension). They show results using 4 m and 15 m resolution RGB+NIR tiles using only 4 dates with significant improvements with respect to other approaches.

Creating classifiers that specifically take into account temporal dynamics for multi-temporal imagery is important, since these dynamics could be lost when training on large amounts of data. In [25], a deep Recurrent Neural Network (RNN, using Long Short Term Memory, LSTM, cells) is used for change detection between two satellite images. Deep RNNs have also been used for land cover classification yielding very good performances with respect to simple temporal stacking [26].

### 1.5. Deep Learning on a High Performance Computing Cluster

Land cover map production at a 10 m resolution on a country-wide scale has large computing and storage requirements. Deep Learning training is highly computationally intensive and benefits from large amounts of data. Moreover, several dozens of passes (epochs) need to be made over the training data. In this work, a HPC cluster infrastructure was available to support this. Yet, efficiently scaling distributed stochastic gradient descent for an HPC architecture while dealing with imbalance in the reference data is a challenge. We address this challenge by proposing a novel data augmentation and distribution across the computational nodes of the cluster.

### 1.6. Contributions

In our work we address the challenges described above—country-scale data with high temporal frequency, sparse annotations across varying climatic regions, imprecise segmentation produced by standard deep CNN approaches—by making these contributions:

1.  a framework of patch generation, data augmentation and patch distribution in a computing cluster that alleviates the very high class imbalance in the dataset;
2.  a novel adaptation of the U-Net model for dealing with sparse data, while producing detailed and fine-grained class predictions;
3.  we show results in an operational setting on the country scale for multi-temporal multispectral production-quality data obtained from the Sentinel-2 constellation.

## 2. Data

The data used in this work were selected in order to assess the performance of the proposed algorithms allowing to draw conclusions for an operational setting. Imagery volume, reference labels quality and quantity, as well as landscape and eco-climatic variability are representative of operational, country-scale land cover map production contexts.

### 2.1. Remote Sensing Imagery

Eleven Sentinel-2 tiles (100 km × 100 km) covering different areas of metropolitan France were selected. They are shown in Figure 1. For each tile, all available Sentinel-2 through the Theia Land Data Centre (http://www.theia-land.fr/en/presentation/products) for the year 2016 were

used. The Sentinel-2 products are orthorectified prior to their release by ESA (Level-1C) and then, Theia processing chains based on the algorithms described in [27] perform cloud (and cloud shadow) screening and atmospheric corrections to produce the Level-2A imagery used in this work. Only the bands with 10 m resolution (blue, green, red, near-infrared) and 20 m resolution (four bands in the red-edge and two in the short-wave infrared) are be used. The 20 m resolution bands are resampled to 10 m using a bicubic interpolation with the appropriate spatial radius to avoid over-smoothing or aliasing.

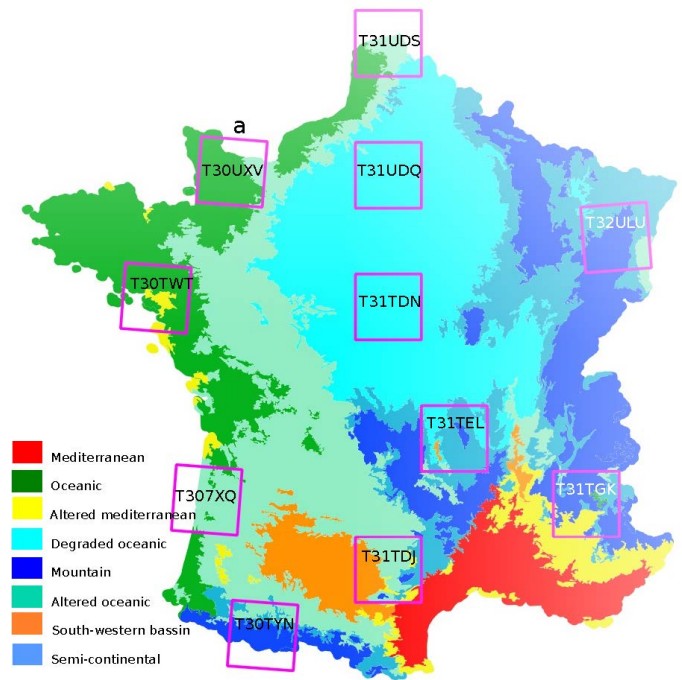

**Figure 1.** Imagery used in this work. Location of the 11 Sentinel-2 tiles and eco-climatic areas.

As described in [8], linear temporal gapfilling is used to estimate image reflectance for pixels flagged as clouds, cloud shadows, saturation or non acquired and a subsequent temporal resampling is applied in order to reconstruct temporal profiles having identical time stamps. This allows to have the same image features for all pixels despite the different orbit cycles (effective acquisition dates), and despite the presence of any artifact such as clouds. Of course, the number of acquisitions for which the land is visible varies across tiles and even within every given tile. It ranges from five clear acquisitions in northern and mountainous areas to more than 30 in the sunny mediterranean areas where two satellite orbits overlap.

At the end of this procedure, each pixel at 10 m resolution is described by 330 features (10 spectral bands for 33 dates).

## 2.2. Reference Data Sources and Nomenclature

The reference dataset is composed of the same existing data sources and produced with the same procedure as described in [8], that is, a fusion of Corine Land Cover (CLC) 2012 [28], the French National Geographic Institute "BD Topo" [29], the agricultural Land Parcel Information System "Registre Parcellaire Graphique" (RPG) [30] and the Randolph Glacier Inventory [31]. Therefore, the nomenclature is composed of the following classes: **CUF** Continuous urban fabric; **DUF** Discontinuous urban fabric; **ICU** Industrial or commercial units; **RSF** Road surfaces; **ASC** Annual summer crops; **AWC** Annual winter crops; **IGL** Intensive grassland; **ORC** Orchards; **VIN** Vineyards; **BLF** Broad-leaved forest; **COF** Coniferous forest; **NGL** Natural grasslands; **WOM**

Woody moorlands; **BDS** Beaches, dunes and sand plains; **BRO** Bare rock; **GPS** Glaciers and perpetual snow; **WAT** Water bodies.

*2.3. Training and Test Data*

As usual in supervised classification, we randomly split the available reference data in 2 disjoint sets, one used for classifier training and the other one used to measure the quality of the classification using standard metrics.

The performance evaluation is made over pixels. Since the reference dataset is composed of polygons where the pixels can be highly correlated (agricultural plots, a forest stand, etc.), we ensure that training and test polygons are different, which avoids that two pixels from the same polygon, and therefore, likely very similar, are used one for training and the other for validation. This avoids optimistic accuracy estimations. For this work, 67% of the data were used for training and 33% for validation. The split was done so that these ratios are enforced for each image tile and class.

Table 1 shows the surface in hectares in the reference dataset for each class and each tile. Since the images have a ground sampling distance of 10 × 10 m, 1 hectare is equivalent to 100 pixels. The variability in terms of class composition between the tiles is related to eco-climatic and landscape variability. For instance, BRO is a rare class in most tiles except for T30TYN and T31TGK where it is one of the most represented classes. Similar behaviours can be observed for other classes and other tiles.

**Table 1.** Areas occupied by the different land cover types in the reference dataset (in hectares) in the 11 tiles.

| | T30TWT | T30TXQ | T30TYN | T30UXV | T31TDJ | T31TDN | T31TEL | T31TGK | T31UDQ | T31UDS | T32ULU | Total |
|---|---|---|---|---|---|---|---|---|---|---|---|---|
| ASC | 5650 | 3306 | 2415 | 6738 | 3700 | 4154 | 3849 | 159 | 3327 | 1951 | 6900 | 42,149 (2.43%) |
| AWC | 5204 | 85 | 572 | 10,235 | 7982 | 22,297 | 5617 | 866 | 32,808 | 9125 | 4511 | 99,302 (5.74%) |
| BLF | 2063 | 2659 | 11,636 | 3152 | 9358 | 27,161 | 10,855 | 8144 | 12,829 | 4465 | 17,184 | 109,506 (6.32%) |
| COF | 2546 | 112,152 | 7544 | 803 | 10,961 | 5594 | 47,692 | 57,027 | 2442 | 94 | 52,772 | 299,627 (17.31%) |
| NGL | 272 | 0 | 18,862 | 0 | 7735 | 62 | 1307 | 51,040 | 24 | 201 | 1484 | 80,987 (4.68%) |
| WOM | 3856 | 3952 | 11,312 | 552 | 3788 | 1433 | 3465 | 18,915 | 749 | 466 | 518 | 49,006 (2.83%) |
| CUF | 368 | 1019 | 203 | 863 | 106 | 157 | 1183 | 20 | 8619 | 906 | 941 | 14,385 (0.83%) |
| DUF | 5962 | 11,108 | 1536 | 3059 | 1501 | 1949 | 7019 | 833 | 31,143 | 5045 | 6514 | 75,669 (4.37%) |
| ICU | 5331 | 7951 | 842 | 3116 | 1172 | 2758 | 4689 | 358 | 25,165 | 6525 | 4647 | 62,554 (3.61%) |
| RSF | 108 | 694 | 112 | 138 | 61 | 179 | 584 | 73 | 2031 | 779 | 615 | 5374 (0.31%) |
| BRO | 0 | 0 | 20,990 | 48 | 27 | 0 | 3 | 53,160 | 0 | 0 | 20 | 74,248 (4.29%) |
| BDS | 140 | 4327 | 0 | 374 | 0 | 107 | 0 | 923 | 0 | 348 | 0 | 6219 (0.35%) |
| WAT | 237,383 | 219,169 | 814 | 125,722 | 2204 | 5045 | 1403 | 2293 | 2567 | 113,073 | 3305 | 712,978 (41.21%) |
| GPS | 0 | 0 | 63 | 0 | 0 | 0 | 0 | 2519 | 0 | 0 | 0 | 2582 (0.14%) |
| IGL | 7595 | 648 | 7891 | 18,446 | 5319 | 5493 | 23,270 | 6165 | 4544 | 2479 | 9458 | 91,308 (5.27%) |
| ORC | 48 | 9 | 2 | 120 | 107 | 55 | 5 | 378 | 144 | 19 | 37 | 924 (0.05%) |
| VIN | 139 | 1620 | 6 | 0 | 1002 | 174 | 25 | 3 | 0 | 0 | 190 | 3159 (0.18%) |
| Total | 276,672 | 368,708 | 84,807 | 173,373 | 55,028 | 76,625 | 110,973 | 202,885 | 126,397 | 145,481 | 109,102 | 1,730,051 |

## 3. Method

In this work, we address the problem of *dense* supervised classification of land cover with many multispectral temporal samples, learned using *sparse annotations*. In many remote sensing applications, labeled data are highly sparse: most of the pixels are not labeled, and labeled areas consist of isolated polygons which may be inside objects without fitting their boundaries. An illustration is presented in Figure 2. To tackle this problem, we propose to use a new variant of the fully convolutional U-Net architecture.

The U-Net [13] network is a classical architecture to address the problem of semantic segmentation. Indeed, it is a fully convolutional segmentation network initially applied to biomedical images. Its symmetrical architecture is able to gather context in the encoding phase and accurately localize features in the decoding phase, as illustrated Figure 3.

Due to the high level of sparsity of the labeled data, this classical network suffers from some limitations in our specific application. In our datasets, annotated polygons are sparse and do not necessarily include the boundaries between classes (e.g., the edge between a cultivated field and a

forest). As confirmed by our initial evaluation of the original U-Net architecture in Section 5, this leads to blurry, soft and erroneous class boundaries when applying the network in a dense prediction setting.

In this section we introduce and discuss FG-UNET our modified (FG for Fine Grained) U-Net architecture that is adapted to our specific problem of land cover classification.

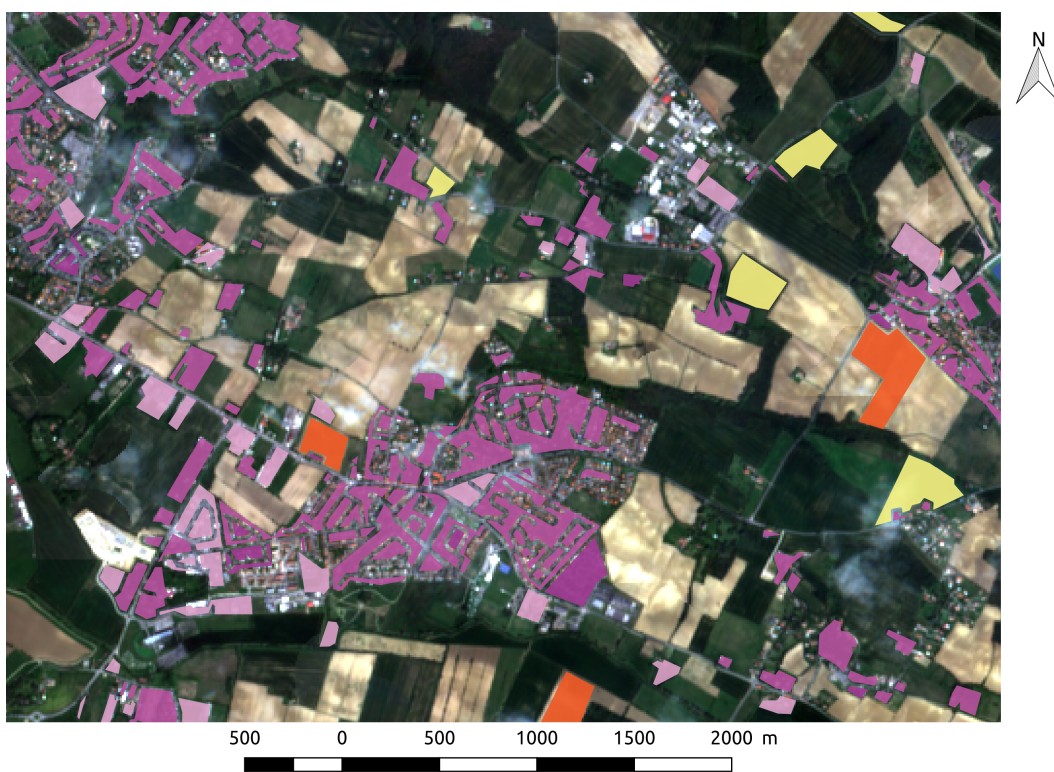

**Figure 2.** An illustration of sparse (non dense) annotations used in this work overlayed on a Sentinel-2 image.

### 3.1. U-Net Architecture Overview

The original U-Net architecture of [13] contains symmetric downsampling and an upsampling paths (5 steps of 2 convolutions, see Figure 3). On the downsampling side, the network takes as input an image patch with $C$ channels of size $N \times N \times C$. For each step, a convolution layer, followed by a ReLU activation function and a subsampling by maximum pooling are used. Therefore, the set of operations of the downsampling path can be written as: `5x [2x [64*2`$^i$` × conv3x3`$_i$`(previous) + relu activation], maxpool2x2]`, where $i \in [0..4]$ are each one of the steps. On the upsampling side, skip connections allow the transfer of raw information from layers on the downsampling side (the crop and copy operations in Figure 3). The upsampling path is implemented through the concatenation of the output of convolution layers on the downsampling side to features produced on upsampled input. The set of operations of the upsampling path can be written using the same type of notation as for the downsampling path: `5x [concat(upsample2x2(previous), output(conv3x3`$_i$`)), 2x [64*2`$^{4-i}$` × conv3x3 + relu activation]]`. The receptive field of the original U-Net architecture is 95 pixels.

For regularization, `Dropout` layers are placed between the two $3 \times 3$ convolutions in each block. The output layer has size $N \times N \times K$ where $K$ is the number of classes.

### 3.2. Fine Grained U-Net Architecture (FG-UNET)

The proposed network, called FG-UNET, is represented in Figure 4. It is inspired from the classical U-Net architecture with the following adaptations:

1. The receptive field and the number of steps in the network are adapted to our remote sensing data resolution and number of dimensions.
2. A pixel-wise fully connected path is added to improve boundary delineation in the produced maps.
3. The temporal component of the data is dealt by replication of the network on subsets of the time series.

These adaptations are described in the following paragraphs.

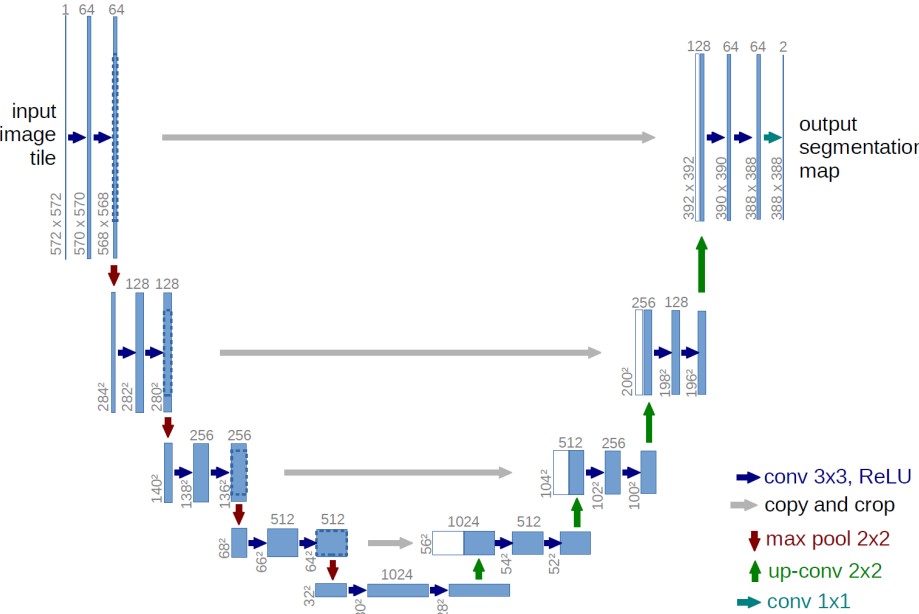

**Figure 3.** The original U-Net architecture [13].

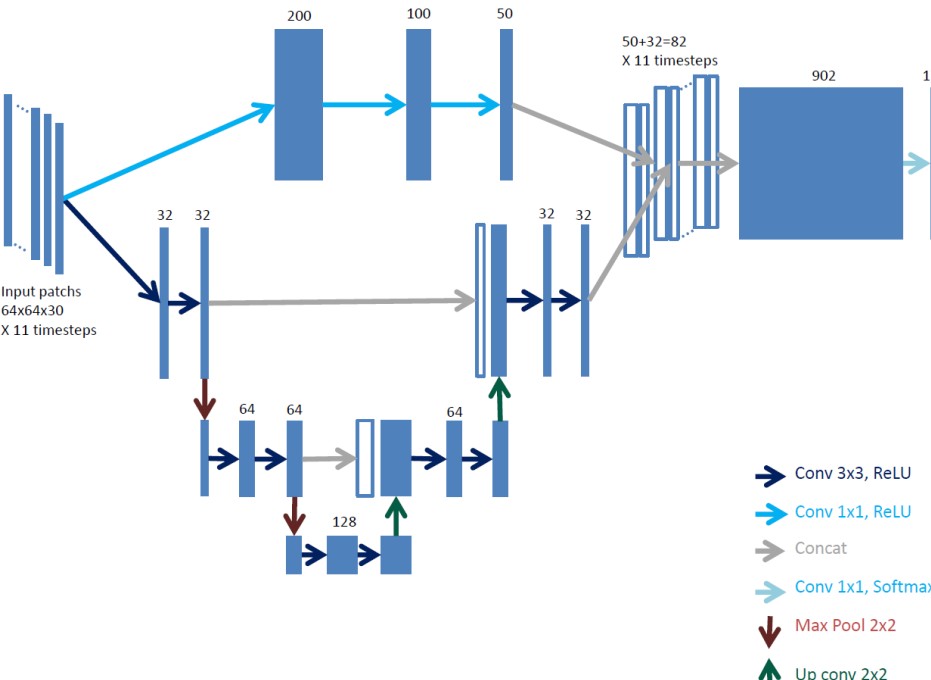

**Figure 4.** The proposed FG-UNET architecture.

### 3.2.1. Receptive Field and Network Size Adaptation

On 10m resolution images, features such as trees, roads and buildings are only a few pixels wide. Moreover, at this resolution, texture details for classes such as vineyards, forests and industrial areas is fine-grained, with sub-pixel frequencies up to several pixels. Thus, the original receptive field is not adapted, and having a 5 step deep network increases the number of parameters that need to be learned. The depth of the original U-Net network was thus unnecessary in this application. A more shallow U-Net, with three steps, was preferred in order to limit overfitting and computation time. The resulting receptive field is 23 pixels. Moreover, we reduced the number of filters, using 32 filters on the first and last layers of the network and adapting the number of filters on the other layers in consequence. Also, instead of retaining only the valid part of the image at the output of a convolution, the whole image is kept and the borders are trimmed at the output of the network. This simplifies the workflow in the prediction step.

### 3.2.2. Pixel-Wise $1 \times 1$ Path

As the state-of-the-art shows, some classes can be very well classified by pixel-wise classifiers. Moreover, during training, the receptive field of the network will cover both non annotated and annotated pixels. When only one or few non-central pixels in the receptive field are annotated, the weights of the network will be modified only based on the gradient produced on those pixels. We anticipate that this will induce smooth and blurry boundaries between classes during inference. To address this issue, and to add a low-parameter count pixel-wise classifier in our network, we propose to add a convolutional $1 \times 1$ pixel-wise classifier in parallel with the convolution based one. Thus, the network is augmented with a cascade of three layers of 200, 100 and 50 $1 \times 1$ convolutional filters whose output is fused with the output of the $3 \times 3$ filters.

### 3.2.3. Time Series Handling

The data used for this study are 33 Sentinel-2 images with 10 spectral channels for the entire year 2016. Two approaches of network design to handle this kind of data are possible: early fusion and late fusion. Early fusion supposes that the first layer takes as input $N \times N$ patches with $C \times T$ channels, where $C = 10$ spectral channels per patch and $T = 33$ temporal samples. Thus, a fusion of all these acquisitions at the beginning of the network would drastically increase the number of parameters in the first layers, where back-propagated gradients have low values and high levels of noise.

The method chosen is a late fusion. The input is broken up into $G$ temporal groups containing $T/G$ temporal samples of $C$ channels. The network takes as input one temporal group and outputs an intermediate representation. This network is duplicated $G$ times, with parameter sharing. These intermediate representations are then concatenated, and input to a linear classification layer to obtain the final prediction.

In our case, $G = 11$ and $T = 33$ so each group has 30 channels. This value for $G$ is a satisfactory trade-off between accuracy and learning time. Figure 4 shows the input layer of the FG-UNET architecture with the 11 temporal groups of 30 channel patches that are formed.

The late fusion approach also enables time-series processing with LSTM layers. We aim to evaluate the performance gain of using a convolutional implementation of LSTM layers, ConvLSTM2D [32], with respect to simple concatenation of the intermediate representations.

### 3.2.4. Sparse and Imbalanced Annotations Handling

In order to reduce the influence of class imbalance, the loss function chosen is weighted categorical cross entropy [33]. The formula for this function, for one patch, is given in Equation (1).

$$\mathcal{L} = \sum_{pixels} \sum_{k=1}^{class\_nb} weight_k \cdot y_{true_k} \cdot \log\left(y_{pred_k}\right) \qquad (1)$$

We encode the ground truth using either one-hot vectors of size $K$ for annotated pixels, or vectors of zeros of size $K$ for unlabeled ones. These vectors are packed into patches of size $N \times N \times K$. This representation is well suited to sparse labels, because unlabeled pixels will have zero loss and are therefore, not taken into account.

### 3.2.5. Spectral Indices

Classical spectral indices can be added as extra channels in the training patches. Some examples are:

- Normalized Difference Vegetation Index (NDVI), computed using Sentinel-2 spectral bands B4 and B8;
- Normalized Difference Water Index (NDWI), computed using Sentinel-2 spectral bands B3 and B8;
- Brightness, as the mean of the ten Sentinel-2 available spectral bands.

We expect that our method and, in general, non-linear classification methods do not benefit in any significant manner from using these indices, since the relations they express can be learned from the data. We perform experiments with and without these indices to verify this hypothesis.

### 3.3. Training Method

To efficiently learn the proposed FG-UNET model, a training method was set up in order to take into account the specificities of satellite images, of the reference ground-truth data and of our HPC infrastructure. Indeed each Sentinel-2 tile used in this study has a size around 80 GB (16 bit encoding, 33 dates, 10 spectral bands, at 10 m resolution over 110 km $\times$ 110 km) which is much bigger than current images used in deep learning applications, labels used are very sparse and class labels are strongly imbalanced. Our HPC cluster provides 252 compute nodes with 24 CPU cores and 120 GB of memory per node, that is, a total of 6048 cores and 30TB of memory.

### 3.3.1. Preprocessing

For faster convergence, CNN models need input data that has normalized values, stored as floating point values. Some works normalize by subtracting the mean and dividing by the standard deviation of each channel. Another approach, used here in order to normalize patches used during learning phase, is to compute the extreme values for each band. We use 3rd and 97th percentiles computed across the whole dataset instead of minimum and maximum values, due to outliers or missing pixels (no data due to orthorectification) potentially present in images. In this way, for each pixel $(i, j) \in N \times N$ of each channel $c \in [0..329]$, each channel $c$ of input patches $p$ is normalized in the interval [0,1]:

$$p'_c(i, j) = clip\left( \frac{p_c(i, j) - 3^{rd} pct(c)}{97^{th} pct(c) - 3^{rd} pct(c)}, 0, 1 \right) \tag{2}$$

Reference data are available as polygons. They are split in train and test sets at the polygon level in order to avoid that two pixels from the same polygon are used one for training and the other for validation. These polygons are projected onto a raster to create class index images of the same size as the satellite image tiles but with only one integer channel containing the class (land cover type) index. A zero index signifies a non annotated pixel.

### 3.3.2. Distributed Patch Generation

Deep learning methods take advantage of large amounts of data better than classical machine learning. Thus, in our case, we aim to learn land cover classification models from multiple tiles. On one hand, extracting and storing all patches from all tiles before the learning phase is not conceivable due to I/O limitations, thus patch generation must be performed on the fly. Yet, due to tile size (80 GB) and

available memory on each node (120 GB), one node cannot load more than one tile. Consequently, tile loading is distributed among several nodes.

As described in Section 2.1, input images are composed of 10 spectral bands and 33 time steps. We take $N = 64$ thus a patch size of $64 \times 64$ pixels is provided to the network, with $C = 330$ bands.

Furthermore, as annotations are sparse, we want to eliminate patches that do not contain sufficient annotated pixels. We set the threshold of annotated pixels at 10%. We precompute the valid patch positions for each tile to speed up patch choice.

At each training iteration we generate a batch of patches. However, some tiles do not contain some of the classes: mountain tiles do not contain beaches and vineyards while tiles covering plains do not contain glaciers. Moreover, coastal tiles contain large amounts of only one class: water. To keep a balance between different classes, patches must be sampled carefully across different tiles.

Our solution is to have a centralized patch sampling mechanism, and  for each node, to extract the patches that correspond to the tile it has loaded and then to communicate these patches to other nodes so that all the nodes have equal workloads. The full algorithm is given in Section 3.3.6.

### 3.3.3. Hyperparameters for Distributed Training

Deep Learning frameworks are efficient in performing computations on large batches of images. Usually the size of the batch is set to the maximum for which the training network graph fits in memory. In our case this is $b = 8$ patches. Distributed training must thus use these batch sizes across all nodes, giving an effective batch size which for our network is $B = \#nodes \times b$.

As remarked in other works [34], training with large batches requires using a learning rate warm up period. We use the Adam optimizer [35] with a base learning rate $lr = 0.001$. For the first $w = 5$ epochs the learning rate is modulated by a warm up factor:

$$lr'_i = \frac{lr}{\#nodes} \left( \frac{\#nodes - 1}{w} i + 1 \right) \tag{3}$$

where $i \in [0..w]$.

### 3.3.4. Class Imbalance Handling

The frequency of the different classes in the Sentinel-2 dataset is heavily imbalanced, showing a ratio of 1–100 between the least and most frequent classes, as illustrated in Table 1. We compute class weights, used in Equation (2) as follows:

$$weight_k = \frac{n_{tot}}{n_k} \tag{4}$$

Furthermore, during training, instead of choosing a random patch, we first randomly choose a class before choosing a random patch for this class. This increases the number of patches containing minority classes that are used for training.

### 3.3.5. Data Augmentation

We use data augmentation to introduce learned invariances in the model. We apply rotation (90°, 180° or 270°) and mirroring (horizontal and vertical). This is particularly useful for classes with very few annotations to avoid always giving the same patches to the network.

### 3.3.6. Complete Batch Creation Procedure

Initially all nodes load one satellite tile. To produce one batch during training the patch generation procedure, executed by all compute nodes, is the following:

1. $B = \#nodes \times b$ examples of a randomly chosen class are sampled
2. $B$ patches containing at least 10% of pixels of the respective class are randomly chosen

3.　the worker $n_i = 0$ (first node):

- determine which nodes have loaded the tile of each of the *B* patches
- sets up the patch transfer scheme between nodes: $t_{ij}$. Say a node $n_i$ must fill $c_i$ patches in the batch. If $c_i > b$, $n_i$ must send $c_i - b$ patches to another node. If $c_i <= b$, $n_i$ must receive $b - c_i$ patches for another node. We set $t_{ij}$ as the number of patches node $n_i$ must send or receive to nodes $j \in [0..\#nodes]$ so that in the end it has *b* patches

4.　each node $n_i$:

- receives the transfer scheme $t_{ij}$
- $\forall j \in [0..\#nodes]$, if $t_{ij} < 0$ node $n_i$ sends $-t_{ij}$ patches to node *j*. Else, it receives $t_{ij}$ patches from node *j*

5.　if spectral indices are enabled, they are computed and additional channels are added to the patches
6.　data augmentation is applied to the patches.

### 3.4. Classification Method

The proposed FG-UNET model is fully convolutionnal. Thus, the classification can be performed on images of any size, as long as they can fit in memory. In this work, we chose to split input Sentinel-2 tiles into patches of size $512 \times 512$ pixels.

However, U-Net classification can present poor performances near border of images. This is due to the fact that padded convolution can introduce a bias in the border of the output image. To overcome this issue, patches are extracted from input tiles with an overlap of 16 pixels.

Once each of these patches is classified, the output classification map of the tile is built.

## 4. Experimental Setup

### 4.1. Experimental Configuration

To test our method, the hardware environment is an HPC infrastructure: a cluster on a low latency network (Infiniband) with shared GPFS storage. Each node has 24 CPU cores and 120 GB of memory. In total, there are 252 compute nodes available for a theoretical maximum of more than 6000 CPU cores. Our software environment is composed of CentOS 7 operating system, Python 3.5 as programming language, the high-level neural networks API Keras with TensorFlow back end, and Horovod for a distributed training. Our code is publicly available online (DOI 10.5281/zenodo.1474224; https://github.com/vpoughon/RT_DL_OSO_public).

### 4.2. Baselines

As mentioned previously, an operational processing chain based on the approach described in [8] is currently used to produce land cover maps. It uses Random Forests (RF), a state-of-the-art remote sensing image classifier. We use the same RF implementation to assess convolutional neural networks (CNN) method. It is provided by OpenCV through the supervised classification framework of Orfeo Toolbox 6.2.

Parameters used are 100 trees in the forest, a maximum depth of the tree of 20 and at least 10 samples in each node. For noise reduction, a regularization of generated map is performed using a structuring element with a radius of one pixel. This RF method is applied to the same dataset as the CNN approach.

We introduce several other baselines:

1.　**MLP 3 layers**: the $1 \times 1$ 3 layer convolutional path of the FG-UNET model by itself. Time series handling is however done in an early fusion manner, the input layer thus having 330 channels.

2. **LSTM 3 layers**: the 1 × 1 3 layer convolutional path of FG-UNET, using input channel grouping (11 groups of 3 channels) and late fusion through an LSTM layer instead of concatenation followed by linear classification.
3. **U-Net 3 steps**: the U-Net part of FG-UNET by itself, with late fusion through concatenation and linear classification.
4. **TSFG-UNET**: We replace convolutions by convolutional LSTM layers.

In Table 2 we compare several useful training quantities:

1. *number of parameters* for each convolutional neural network tested. CNN parameter numbers can not be compared with RF, whose parameters are chosen in a discrete set of values.
2. *learning times* for a learning performed on 11 nodes containing 24 CPUs each
3. *learning time without parallel computing*. For CNN methods it corresponds to the previous column multiplied by 11 nodes and 24 CPUs. It can be compared to RF learning which is directly performed on one CPU, because the implementation used does not allow parallelism, but other implementations could be used in order to reduce the time, since each tree in the forest can be trained in parallel.

CNN methods are trained for 50 epochs, a value chosen experimentally, beyond which the loss did not decrease further.

The ability to work on a CPU HPC cluster is important for the application. However, a training was performed on a GPU for comparison. For the FG-UNET method, the learning time was 8 h on a GPU.

**Table 2.** Parameter count and training time comparison.

| Method | Parameters | Learning Time | Learning Time on 1 CPU |
|--------|-----------|---------------|------------------------|
| RF | - | 25 h | 25 h |
| MLP 3 layers | 40,717 | 7 h | 1760 h |
| LSTM 3 layers | 324,024 | 140 h | 36,960 h |
| U-Net 3 steps | 485,297 | 7 h | 1760 h |
| FG-UNET | 525,997 | 13 h | 3300 h |
| TSFG-UNET | 4,137,272 | 237 h | 62,700 h |

*4.3. Metrics and Their Limitations*

The statistical comparison presented here uses standard metrics: Cohen's Kappa coefficient as an overall measure and the F-Score coefficient (the harmonic mean between precision and recall). Although the use of Cohen's Kappa can be argued when comparing different datasets [36] we use it here to compare different methods on the same datasets.

However, in the operational context, these metrics have some limitations. In the validation dataset used in these experiments (similar to [8]), pixels located in the center of spatially coherent areas such as agricultural plots or city blocks are far more common than pixels on the edges or in the corners. This implies that global scores like Overall Accuracy, Kappa, F-scores do not fully reflect the quality of the classification. Indeed, the degradation of minority elements like edges and corners can be overshadowed by an improvement in the central areas. However, these negative effects are very visible in the output maps. For example, in Figure 8a, the fine details of the harbour area are smoothed out. The aim of this section is to show how this degradation can be quantified through a robust and fast measure that evaluates whether or not a contextual classification method respects the high spatial frequency areas in the image. If a dense testing dataset is available, validation samples can be split into separate categories that focus on the corners, edges, or fine elements, as is done

in [37]. However, in this study, the validation dataset is sparse, meaning that such an approach can not be used. Furthermore, the reference polygons are eroded to avoid errors due to misregistrations between satellite images and annotations. The idea is, therefore, to use a pixel-based classification as a validation map, under the assumption that pixel-based methods capture the geometry of high spatial frequency areas of the image. The pixel-based classification map has other issues, such as noise and the incorrect characterization of context dependent classes, but it should better preserve the corners and fine elements in the image. Therefore, it is be used as a base to validate the quality of the geometry of a classification map generated by a contextual method.

The most visible deformation source is the rounding of sharp corners in the image, as shown in Figure 8, so the geometrical quality measurement is be based on the preservation of corners in a contextual classification, with respect to a pixel-based classification. This can be achieved by extracting the corners respectively in the pixel-based and contextual classification results, and by seeing how many of these corners match up within a certain spatial radius. The process for extracting the corners in a classification map is as follows. First, the classification output is split into binary maps for each class. Then, a Line Segment Detector, based on the work in [38], generates a set of segments that follow portions of straight lines in the binary maps. In order to find corners between areas of various classes, all of the class segments are then merged. A corner is found if the extremity of one segment is close enough to another segment, and if their angle is smaller than a given threshold.

Once the corners are extracted, the geometrical resemblance of two classification maps can be quantified as the ratio between the number of corners that appear in both images to the number of corners in the reference image. To obtain a robust measurement, ten reference classifications were used, generated by using different subsets of the training data (cross validation). This was done for a pixel-based classification using Random Forest, and for the CNN result.

Results of this geometric precision analysis are presented in Section 5.4.

## 5. Results

In order to assess the proposed FG-UNET network, various experiments are carried out (Figure 5). First, it is compared to the Random Forest baseline. Second, the importance of individual items of the proposed architecture is evaluated. These experiments contain:

- statistical comparison on tiles and on individual classes;
- visual comparison on level of detail of the produced maps.

### 5.1. FG-UNET vs. RF

Statistical results represented by Cohen's Kappa coefficient are represented for each tile on Figure 6. It shows that FG-UNET method has better performances than RF for all tiles except T30TYN and the largest difference is for T31UDQ. This can be explained by the different distribution of classes in the different tiles. T31UDQ is centered over Paris and the proportion of urban areas is the highest of all tiles. Conversely, T30TYN has few urban areas and natural grasslands (NGL) is one of the most present classes and, as it will be shown later, RF yields a high accuracy for this class.

Detailed results for each class and for tile T30TWT are represented on Figure 7. This tile is chosen because of the good balance between the classes (see Table 1), allowing a richer analysis. We observe that:

- FG-UNET is similar or better than RF for all classes, except NGL, a class for which context is less discriminant than spectral and temporal information.
- FG-UNET is significantly better for all urban classes (CUF, DUF, ICU, RSF). FG-UNET takes advantage of the use of the texture. Contrary to pixel-based methods like RF, convolutionnal networks can take into account the context of the pixel to classify, which is crucial in artificial areas.
- Error bars show that the variability of results is higher for FG-UNET than for RF. FG-UNET and RF learning and classification were performed five times, in order to analyze variability of

models. The Kappa coefficient remains the same, but for some classes FG-UNET results may vary. This variation is mainly found for minority classes, for which the data augmentation used for training is not sufficient to capture the intra-class variability and therefore, the sensitivity to initialization is important.

It is interesting to notice in Figure 6 that the more a tile contains artificial areas (see Table 1), the better are FG-UNET results. Indeed, tiles T30TYN and T31TGK are located on mountain areas, and T31UDQ (bigger gap between FG-UNET and RF) is located over Paris. This is due to the fact that contextual information (texture, mix of materials) is more important for urban classes than for vegetation, for which the temporal and spectral signatures convey more information.

Figure 8 allows a visual comparison on extracts of produced land cover maps (for tile T30TWT). These extracts have a size of 400 × 500 pixels (4 km × 5 km). The legend is presented on Figure 5. It highlights that:

- Road recall is better with FG-UNET, but with a lower precision
- Level of detail is similar with FG-UNET and RF in rural regions
- Comparison of Figure 8a,b highlights that city streets are not classified as roads with FG-UNET, contrary to RF. These streets are not part of the training set, so the expected class is not well defined. However, if some street training samples were part of classes CUF or DUF, FG-UNET with the use of context could be able to classify them correctly, whereas with a pixel-based method like RF, it would be more challenging. On the other hand, these two figures also show that the FG-UNET has more difficulty in preserving sharp details as the harbour area.

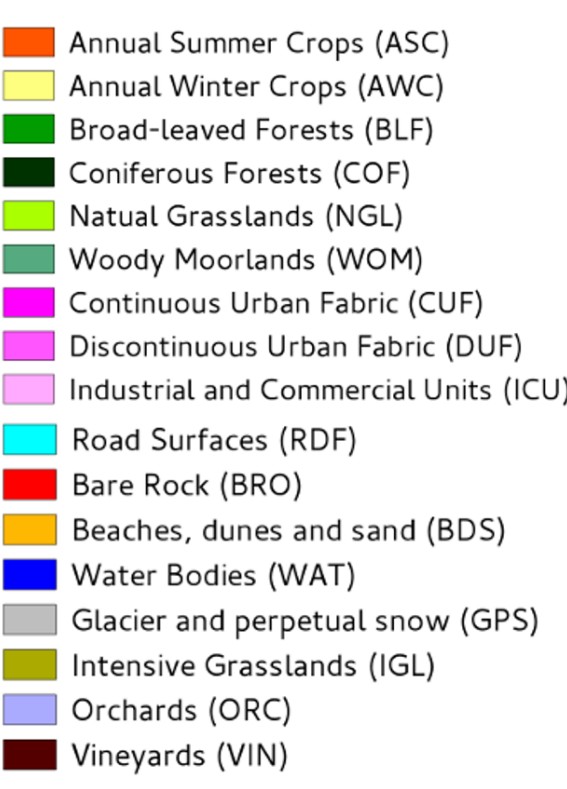

**Figure 5.** Colours of the 17 classes.

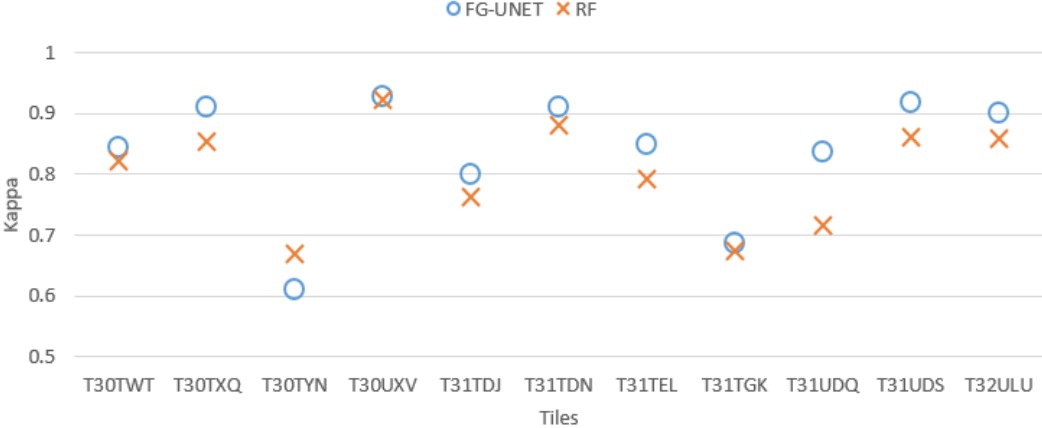

**Figure 6.** Cohen's kappa comparison for all tiles between FG-UNET and RF.

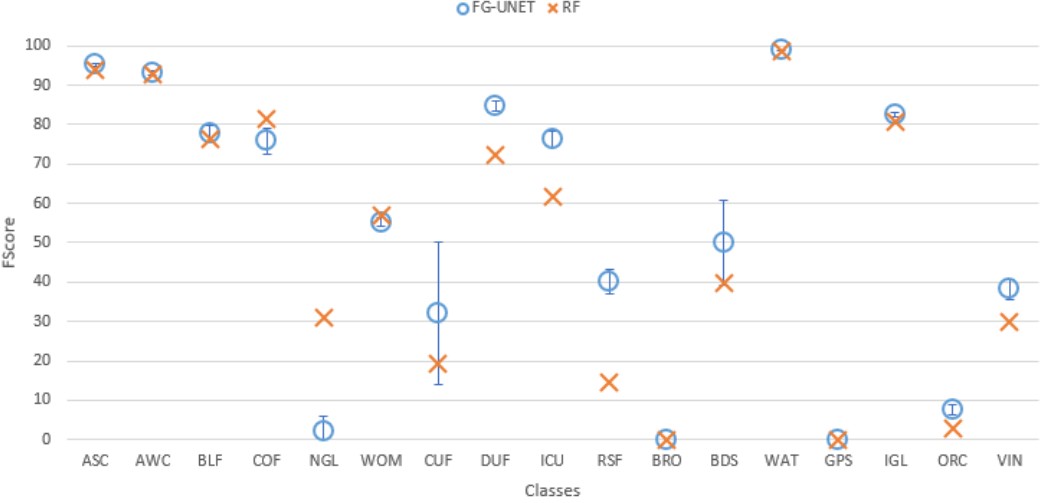

**Figure 7.** FScore comparison between FG-UNET and RF for tile T30TWT.

*5.2. Importance of Individual FG-UNET Features*

In order to prove that the proposed FG-UNET is appropriate for this study, tests of alternative networks close to FG-UNET are performed.

1. FG-UNET vs UNET. This test was carried out in order to prove the importance of the pixel-wise $1 \times 1$ path added to UNET to obtain FG-UNET. Figure 9 shows that FG-UNET gives results slightly above UNET for all tiles. Results are even significantly better for mountain tiles (T30TYN and T31TGK). This is the same phenomenon observed when comparing FG-UNET and RF: pixel-based classification works better for this areas and FG-UNET adds a pixel-based path which complements the contextual classification. The detail by class for T30TWT on Figure 10 shows that FG-UNET is better than UNET (except for CUF). Thus, these statistical results prove the importance of pixel-wise $1 \times 1$ path contribution. Moreover, visual analysis of produced land cover maps (Figure 11) highlights the increasing in level of detail with FG-UNET. For instance, the jetty of Figure 11c is better rendered in Figure 11a than in Figure 11b.

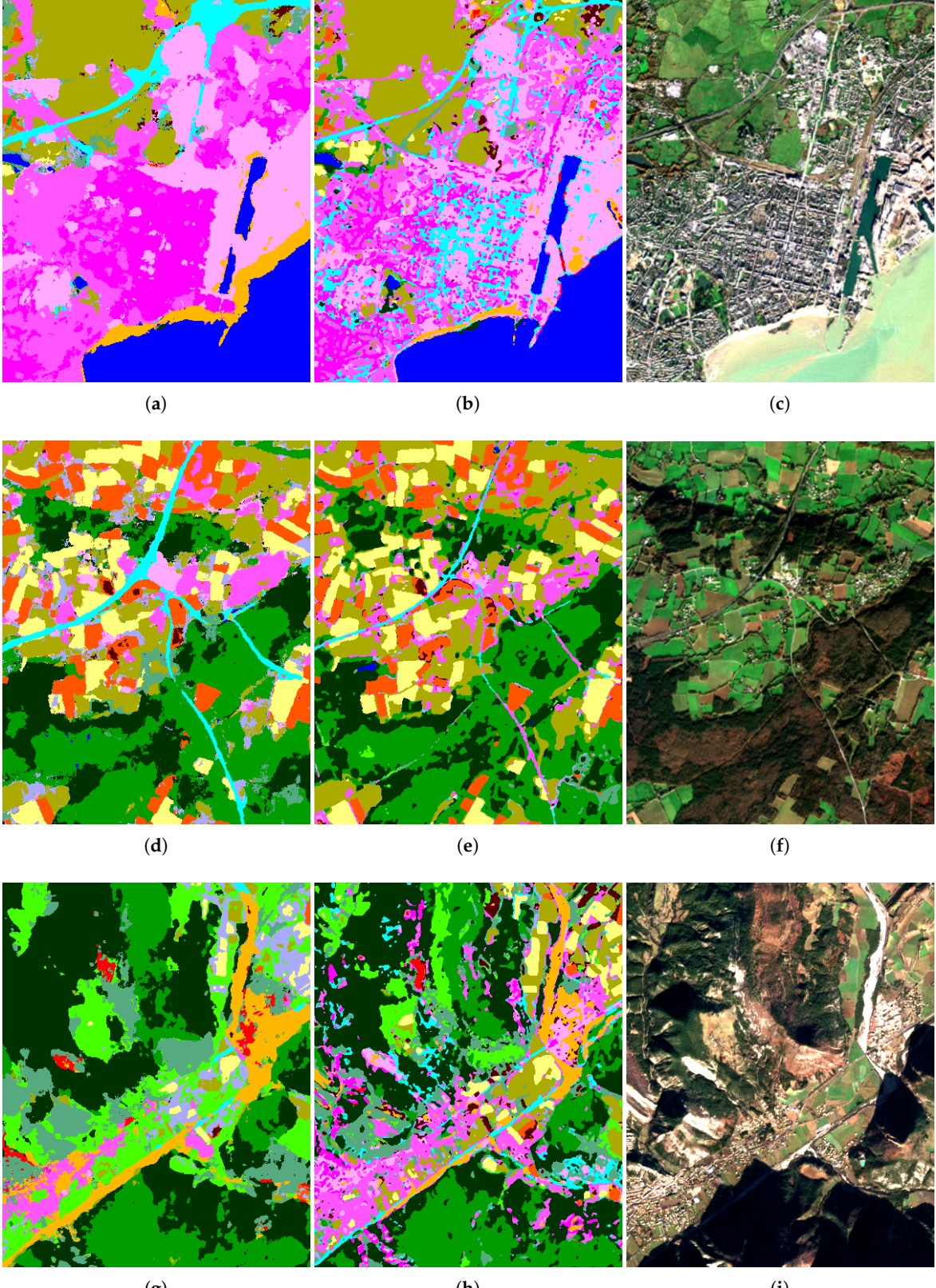

**Figure 8.** Extracts of produced land cover maps with FG-UNET and RF. Top row: an urban area (shades of pink and roads in cyan). Middle row: a mix of croplands (yellow and orange), urban (shades of pink) and forests (shades of green). Bottom row: mountainous area with grasslands, moorlands and forests (shades of green) and some urban settlements. See Figure 5 for the detailed legend. (**a**) FG-UNET; (**b**) RF; (**c**) Image; (**d**) FG-UNET; (**e**) RF; (**f**) Image; (**g**) FG-UNET; (**h**) RF; (**i**) Image.

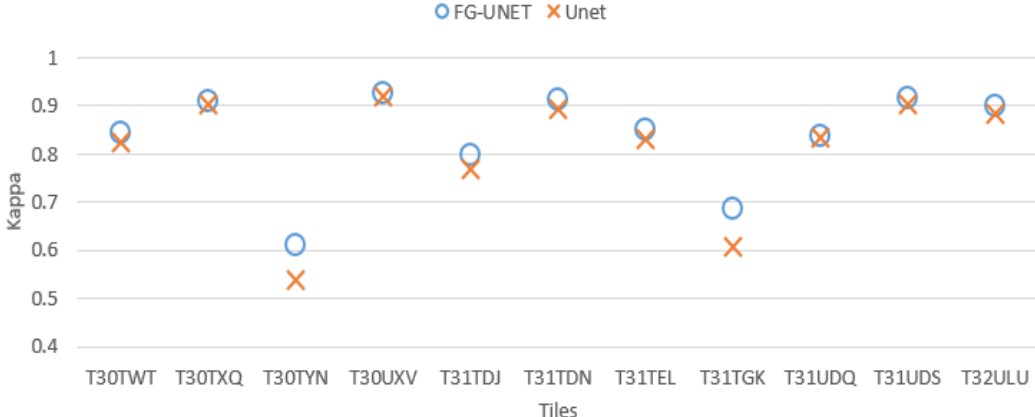

**Figure 9.** Cohen's kappa comparison for all tiles between FG-UNET and UNET.

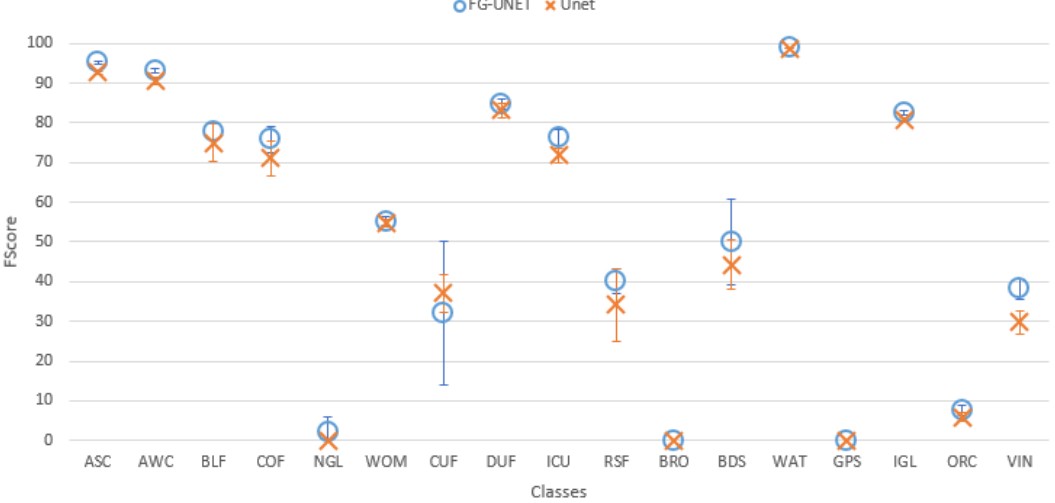

**Figure 10.** FScore comparison between FG-UNET and UNET for tile T30TWT.

2.  FG-UNET vs MLP 3 layers. In this test we compare FG-UNET with a network composed of only the pixel-wise $1 \times 1$ path (MLP). It is a pixel-based method, without use of context, like RF. Results of MLP are below FG-UNET for all tiles, and all classes on T30TWT as represented on Figures 12 and 13. We can also observe that RF provides better results than MLP with 3 layers.

3.  FG-UNET vs LSTM 3 layers. If the temporal aspect is taken into account with a LSTM network, results are better than with a MLP 3 layers. It is still a pixel-based approach, and results are very closed to RF baseline, and consequently below FG-UNET, as shown in Figures 14 and 15.

4.  FG-UNET vs TSFG-UNET. A test was performed replacing all convolutions of FG-UNET by ConvLSTM2D [32], which is a convolutional implementation of LSTM. Figures 16 and 17 show that results of TSFG-UNET are very close to FG-UNET. Yet, as represented in Table 2 the training time is 19 times higher with ConvLSTM2D. Consequently FG-UNET remains more interesting for this application.

5.  Spectral indices usefulness. To assess the usefulness of spectral indices NDVI, NDWI and brightness defined in Section 3.2.5, a test was performed on the tile T30TWT. We can see on Figure 18 that results for all classes are very close with and without the use of spectral indices. It confirms that information brought by these features can be learned from the data.

**Figure 11.** Extracts of produced land cover maps with FG-UNET and UNET. (**a**) FG-UNET; (**b**) UNET; (**c**) Image; (**d**) FG-UNET; (**e**) UNET; (**f**) Image.

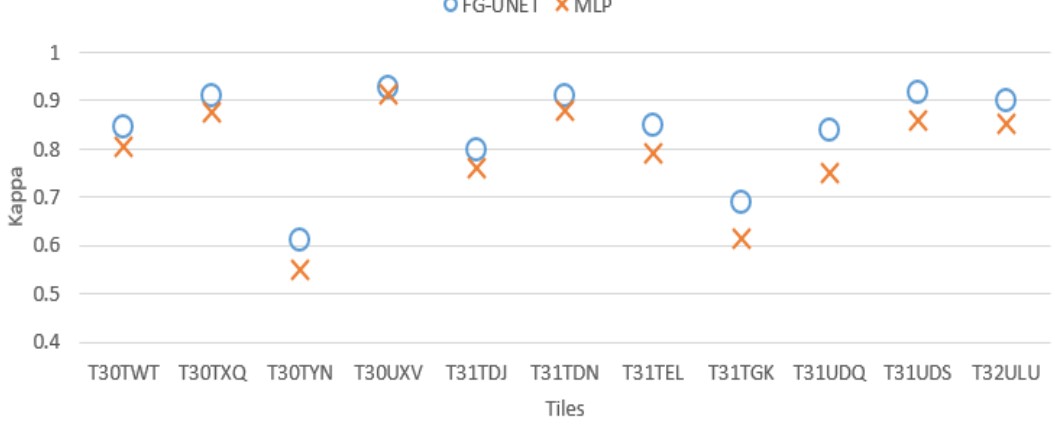

**Figure 12.** Cohen's kappa comparison for all tiles between FG-UNET and MLP.

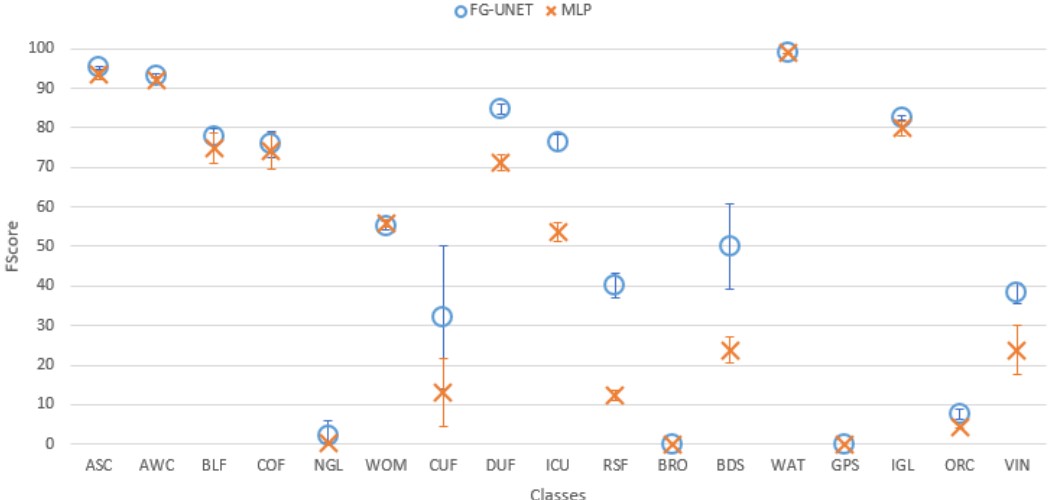

**Figure 13.** FScore comparison between FG-UNET and MLP for tile T30TWT.

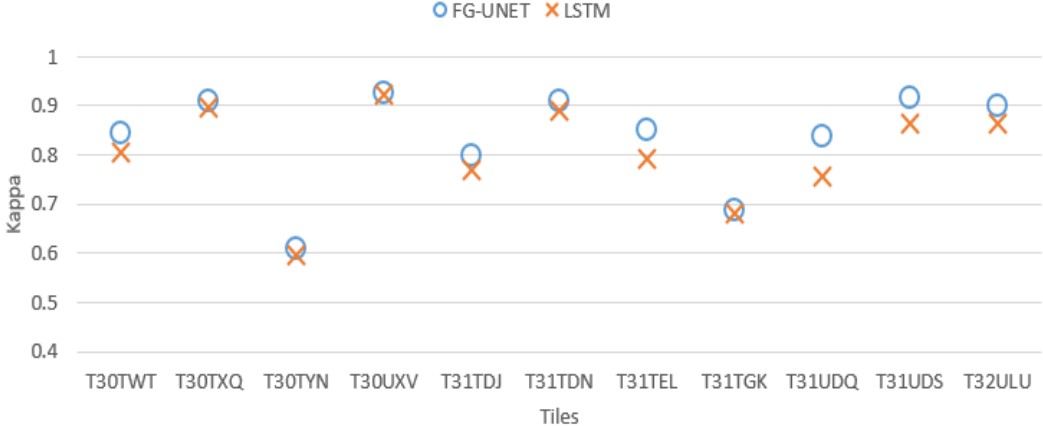

**Figure 14.** Cohen's kappa comparison for all tiles between FG-UNET and LSTM.

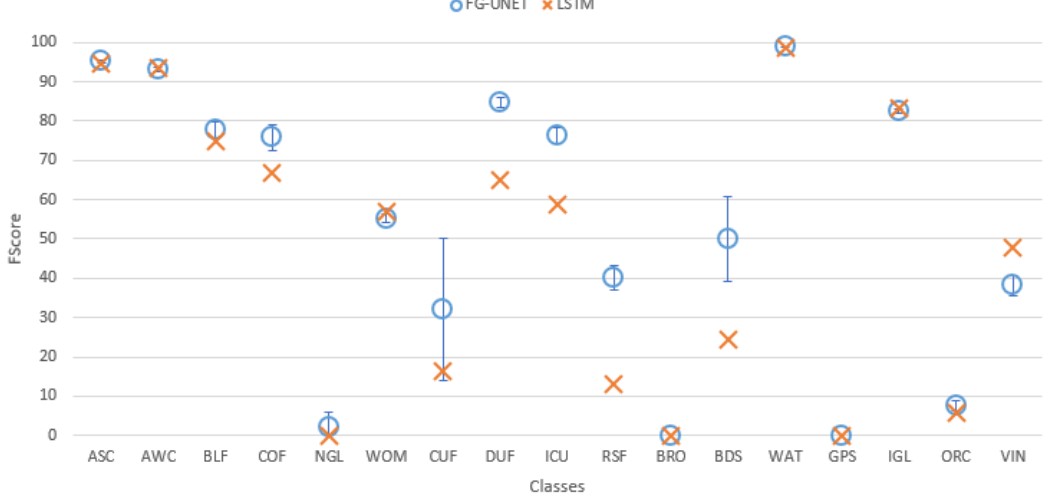

**Figure 15.** FScore comparison between FG-UNET and LSTM for tile T30TWT.

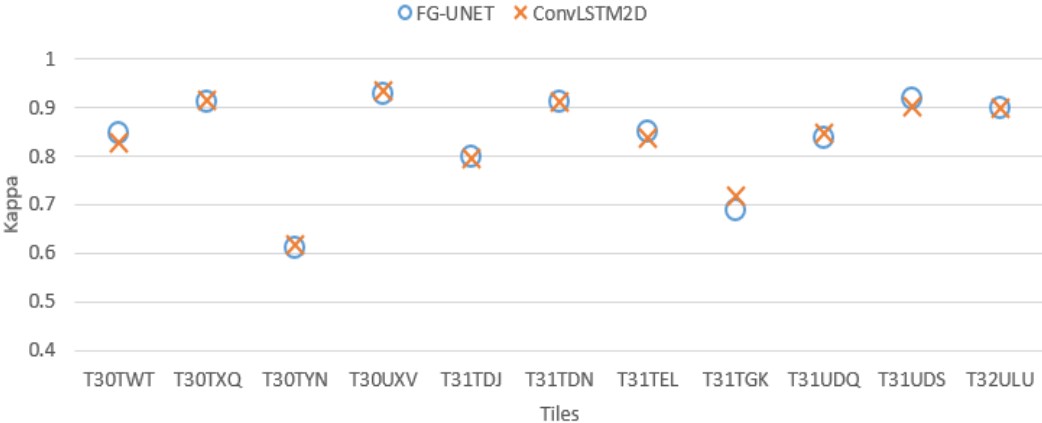

**Figure 16.** Cohen's kappa comparison for all tiles between FG-UNET and ConvLSTM2D.

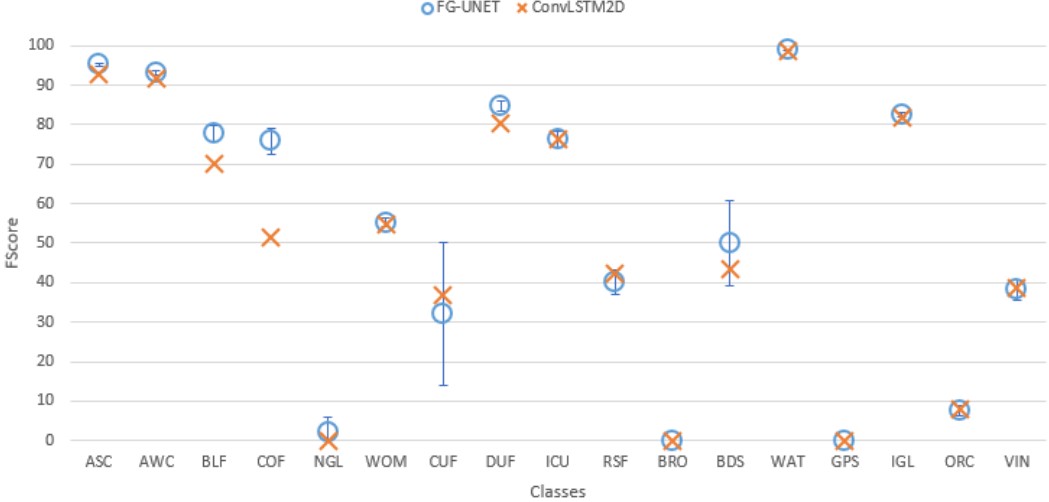

**Figure 17.** FScore comparison between FG-UNET and ConvLSTM2D for tile T30TWT.

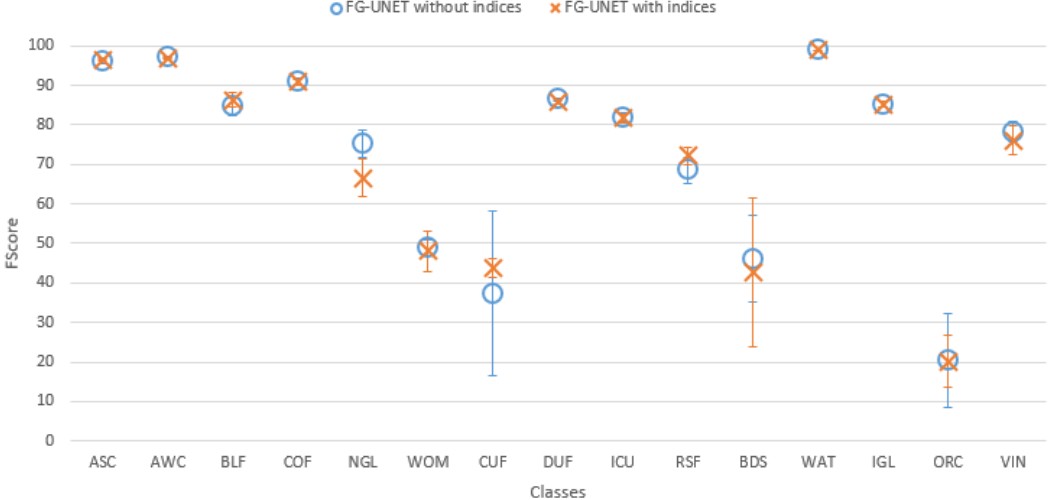

**Figure 18.** FScore comparison on tile T30TWT for FG-UNET with and without indices NDVI, NDWI and brightness.

### 5.3. Cross Domain Transferability

It is interesting to compare the ability of RF and FG-UNET to classify a tile whose training dataset is not used for learning. To do so, for both RF and FG-UNET methods, learning is performed on training datasets of 10 tiles, and the classification is evaluated on the testing dataset of the 11th tile. Results are presented on Figure 19 and compared to the classical results (model learned on training dataset of 11 tiles). We observe on Figure 19:

- a drop in performance when a tile is not part of learning. Yet, it highly depends on the tile content. This drop is particularly high for mountain tiles (T30TYN and T31TGK) whose content is different from other tiles, and for T30TXQ which also has very few AWC labels. On the other hand, their is almost no drop for tile T30UXV.
- FG-UNET suffers from a higher decrease than RF. It tends to show that our implementation of FG-UNET cannot generalize as well as RF.

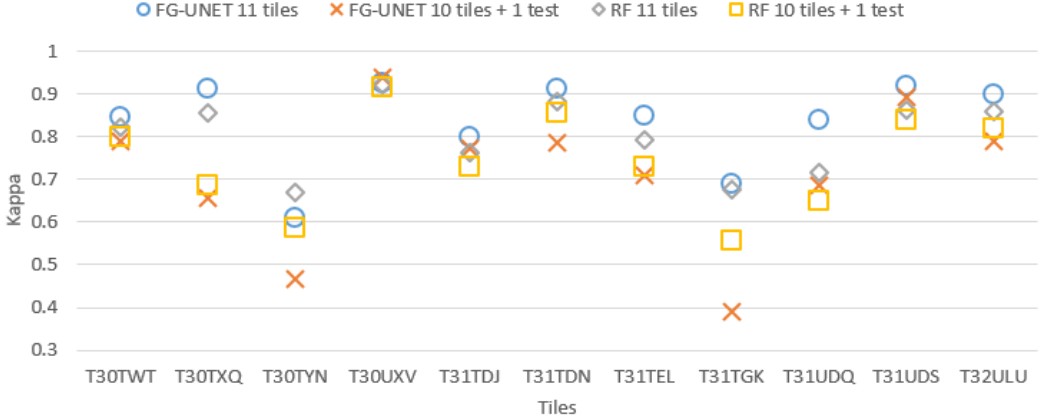

**Figure 19.** Cohen's kappa comparison for all tiles between test of a model learned on 11 tiles and test on a 11th tiles of a model learned on 10 tiles.

### 5.4. Geometric Precision

Figure 20 shows the results of the geometric precision ratio as defined in Section 4.3, averaged over the ten classifications, along with the standard deviation of the measurement. On all of the different tiles, the geometrical accuracy is lower with the FG-Unet than with a pixel-based classifier, Random Forest. It is worth noting that even between pixel-based classifications fewer than 50% of the corners are matched, and that for some tiles, this figure can be lower than 30%. Across all tiles the FG-UNET classification matches 1/3 of the corners matched between the pixel-based classifications. This is coherent with the smoothing observed in the result maps. Furthermore, this is a recurrent issue, as it appears in different geographical areas, and for different reference images.

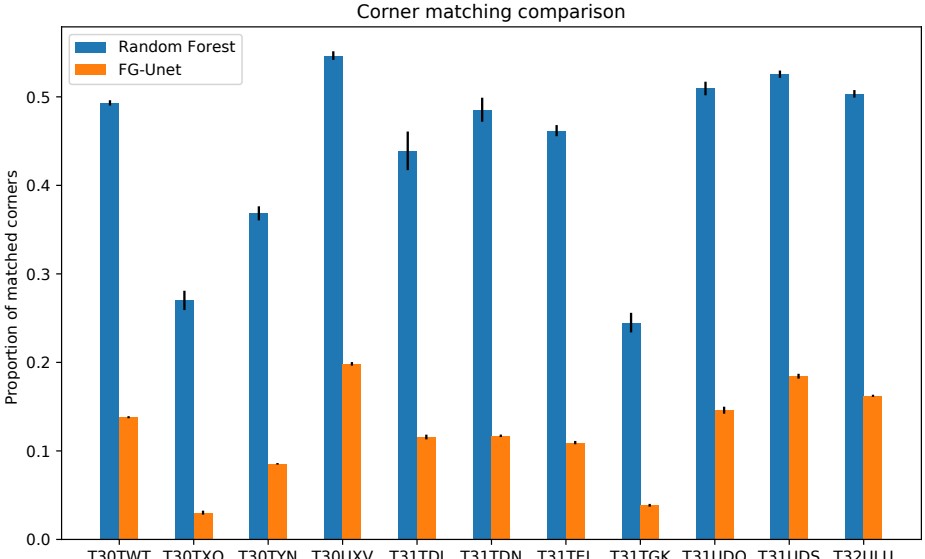

**Figure 20.** Comparison between geometrical accuracy of a pixel level classification map and a map generated by an FG-Unet. Each one was compared to ten different pixel level classifications, in terms of how well the sharp corners matched. The bars show the average and standard deviation of these measurements. The geometrical accuracy for our FG-Unet result is clearly lower than a pixel-based classifier. This means that there is some deformation in the sharp corners.

## 6. Discussion

In light of these results, we can say that the modified U-NET architecture is able to produce land cover maps wich improve the pixel-based classification produced by RF for classes where the pixel context is important, while yielding more geometrically precise results than the original U-NET. This general statement has to be modulated depending on the classes and the types of landscapes. The RF approach seems to better generalize to areas not used for the training and the CNN approach shows higher sensitivity to initial conditions on the training. This deffect could be overcome by locally performing fine-tuning of the CNN model and therefore, reducing the training cost.

In an operational setting, where maps need to be produced at fixed recurring times (for example every year), computational costs are very important issues. The computation times shown in Table 2 are obtained using 11 tiles. If we wanted to map the European Union, we would need 450 tiles, and therefore, the training time would increase by a factor of more than 40. In this context, the ability to fine-tune a pre-existing model is very useful. Indeed, climate differences between years don't allow to use a previoulsy trained model on newly acquired data [39]. While training a CNN model as FG-UNET for every map to be produced is too costly, fine-tuning the model has a cost comparable to the one of training a simpler classifier as RF. In the same way, using the already trained model as a (contextual) feature extractor and replacing the final fully connected layers by a more robust classifier as RF can also allow to combine the advantages of both approaches.

Going further along this path, it may be interesting to revisit existing works on the equivalence between CNN and auto-context random forests [40] in order to find a better trade-off between the ability to automatically extract contextual features and the computational complexity acceptable in an operational map production system.

## 7. Conclusions

In this work, we studied how to use a fully convolutional deep learning architecture to classify Sentinel-2 image time-series at the country scale. Our model, FG-Unet, when trained on all tiles, obtains equivalent or better accuracy than methods that are currently used in production, especially over highly textured areas such as urban classes. However, this improvement comes with a large

computational cost. Without significant engineering effort to reduce the time required to train the model, it is still unclear whether the trade-off is justifiable in a operational context.

This work also confirms the major importance of the quality of the training data. It has a large and measurable effect on the quality of the final land cover map produced.

Nevertheless, none of those challenges are difficult to solve. Future work following this study could focus, firstly, on reducing over-fitting and increasing generalization across different eco-climatic regions. One approach could be to optimise the architecture of the network through automatic methods such as Differentiable Architecture Search [41]. Secondly, to alleviate the problems of sparse annotations, semi-supervised learning methods such as [42] could be employed.

Furthermore, a more fair comparison with Random Forest should be performed. CNNs naturally capture a pixel spatial context due to their wide receptive field. However, random forest is only ever given a single pixel time-series as input. This makes the theoretical comparison of their classification performance slightly biased, because the two classifiers are not given the same information per data point. Nonetheless, the comparison presented in this work remains valid as an empirical investigation of the capacities of the two systems in an operational context.

Finally, our code should be benchmarked on a GPU cluster, which would improve training times, although at a high acquisition cost. Our code is available under an open-source license online (DOI 10.5281/zenodo.1474224; https://github.com/vpoughon/RT_DL_OSO_public).

**Author Contributions:** Conceptualization, A.S., V.P. (Vincent Poulain), J.I. and V.P. (Victor Poughon); Data curation, J.I.; Formal analysis, A.S. and V.P. (Vincent Poulain); Methodology, A.S., J.I., V.P. (Victor Poughon) and D.D.; Project administration, V.P. (Victor Poughon); Software, A.S. and V.P. (Vincent Poulain); Validation, V.P. (Vincent Poulain) and D.D.; Writing—review & editing, A.S., V.P. (Vincent Poulain), J.I., V.P. (Victor Poughon) and D.D.

**Funding:** This research received no external funding.

**Conflicts of Interest:** The authors declare no conflict of interest.

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
