# Peer review of "Land Cover Maps Production with High Resolution Satellite Image Time Series and Convolutional Neural Networks: Adaptations and Limits for Operational Systems"

_remotesensing, doi:10.3390/rs11171986_

Round 1
Reviewer 1 Report
Overall: I believe the manuscript is an important contribution towards the ever-important topic of creating operational architectures for large scale LC-LU mapping. Indeed, although there have been extensive developments on DL architectures (CNN and FCN variants), most of them can be very complex and perhaps non-operational for systematic classification of large areas. The authors make a step forward in that regard, stressing that computational times and simplicity can be just as important as accuracy and combine this with the issue of sparse training labels which are the norm in remote sensing applications. The effort to find a proper trade-off between computational and classifying performance is indeed, commendable. They indeed, raise a valid point of discussion within the remote sensing community for future work and projects alike.
1. Introduction
I think the authors could advance a bit more on referencing some more of the SoA CNN and FCN developments in the past 2-3 years in remote sensing and slowly built into the point that they are trying to make – that a lot of them can be too complex and computationally inefficient for operational use unless we devote an unnecessary large amount of resources (which should be avoided, especially in an era which we should be careful with out energy needs as a human society most of all, and then as EO scientists). Moreover, a lot of these SoA methods are tested on reference datasets with dense training data (ie Vaihingen) which is not what EO scientists usually have to deal in LC mapping but rather sparse labels.
On the topic of fuzzy boundaries, on top of [A*18-19], there have been proposed approaches that refine the predictions using simple GEOBIA, rather than developing overly complex loss functions within the CNN networks [R**1,2].
2. Methods
I believe the readers would benefit from a figure showing an example of the sparsely annotated training samples to appreciate why this work is tackling an important topic (dense prediction from non-dense training samples).
Table 2: With respect to the RF training, there are libraries (such as ‘ranger’ in R) which distribute the training on a given number of nodes. In this case, I would assume that the training of RF would not take more than an hour, so it would be important to stress that you are using a non-parallelized RF implementation with a small number of trees (by all means, 100 trees is probably set for computationally efficiency but not optimal for accuracy).
3. Results
Figures can be hard to follow so a table summarizing the classification accuracy results with numbers would be appreciated (you can even have it as supplementary material).
4. Discussion
I believe there has to be some in-depth discussion towards what a good solution should be for an operational large-scale framework. What would be ideas to further increase the operationability of a DL method? Perhaps using PCA components instead of all images or doing some preliminary feature elimination prior to starting with the training. In the past years, there has been significant work towards parsimony in supervised machine learning classifiers such as RF which have been shown to produce classifying models with maximal accuracy with only a few but discriminate hand-crafted features, almost eliminating training time and decreasing workloads [R3]. In the authors case, it seems that a SoA machine learning implementation (ie., XgBoost instead of RF) with parallelization and a good feature engineering could produce similar results with a minimal computational effort than the range of CNN’s tested here. I could stress that future efforts will perhaps combine the 2 approaches – a supervised ML to produce fast and acceptable dense predictions and an FCN to train on these predictions to further improve and to justify its use.
*Author references
**Reviewer references
References
[1] Fu, Tengyu, et al. "Using convolutional neural network to identify irregular segmentation objects from very high-resolution remote sensing imagery." Journal of Applied Remote Sensing 12.2 (2018): 025010.
[2] Georganos, S., Grippa, T., Vanhuysse, S., Lennert, M., Shimoni, M., Kalogirou, S., & Wolff, E. (2018). Less is more: Optimizing classification performance through feature selection in a very-high-resolution remote sensing object-based urban application. GIScience & remote sensing, 55(2), 221-242.
Author Response
━━━━━━━━━━━━━━━━━━━━━━━━━
RESPONSES TO REVIEWER 1
━━━━━━━━━━━━━━━━━━━━━━━━━
1 Comments and Suggestions for Authors
══════════════════════════════════════
Overall: I believe the manuscript is an important contribution towards
the ever-important topic of creating operational architectures for
large scale LC-LU mapping. Indeed, although there have been extensive
developments on DL architectures (CNN and FCN variants), most of them
can be very complex and perhaps non-operational for systematic
classification of large areas. The authors make a step forward in that
regard, stressing that computational times and simplicity can be just
as important as accuracy and combine this with the issue of sparse
training labels which are the norm in remote sensing applications. The
effort to find a proper trade-off between computational and
classifying performance is indeed, commendable. They indeed, raise a
valid point of discussion within the remote sensing community for
future work and projects alike.
1.1 Answer
──────────
We thank the anonymous reviewer for these comments which confirm the
interest of our work.
2 Introduction
══════════════
I think the authors could advance a bit more on referencing some more
of the SoA CNN and FCN developments in the past 2-3 years in remote
sensing and slowly built into the point that they are trying to make –
that a lot of them can be too complex and computationally inefficient
for operational use unless we devote an unnecessary large amount of
resources (which should be avoided, especially in an era which we
should be careful with out energy needs as a human society most of
all, and then as EO scientists). Moreover, a lot of these SoA methods
are tested on reference datasets with dense training data (ie
Vaihingen) which is not what EO scientists usually have to deal in LC
mapping but rather sparse labels.
On the topic of fuzzy boundaries, on top of [A*18-19], there have been
proposed approaches that refine the predictions using simple GEOBIA,
rather than developing overly complex loss functions within the CNN
networks [R**1,2].
2.1 Answer
──────────
We have completed our reference to Zhu, X.X.; Tuia, D et al. paper
with a short discussion about complexity as suggested by the reviewer.
We have also added [R*1] to show that simpler approaches to boundary
enhancement are possible.
3 Methods
═════════
I believe the readers would benefit from a figure showing an example
of the sparsely annotated training samples to appreciate why this work
is tackling an important topic (dense prediction from non-dense
training samples).
Table 2: With respect to the RF training, there are libraries (such as
‘ranger’ in R) which distribute the training on a given number of
nodes. In this case, I would assume that the training of RF would not
take more than an hour, so it would be important to stress that you
are using a non-parallelized RF implementation with a small number of
trees (by all means, 100 trees is probably set for computationally
efficiency but not optimal for accuracy).
3.1 Answer
──────────
A figure illustrating the sparse annotations has been added.
Section 4.1, where we said that the RF implementation used is not
parallel, we stress that other implementations could be used in order
to reduce the computation time.
4 Results
═════════
Figures can be hard to follow so a table summarizing the
classification accuracy results with numbers would be appreciated (you
can even have it as supplementary material).
4.1 Answer
──────────
This was a difficult choice for the authors: presenting results in a
tabular form or in a graphical one. The tabular form, although
containing precise information, was even more difficult to interpret
than the graphical one, where it is easier to appreciate similarity
between tiles or classes and even confidence intervals and variability
when pertinent.
5 Discussion
════════════
I believe there has to be some in-depth discussion towards what a good
solution should be for an operational large-scale framework. What
would be ideas to further increase the operationability of a DL
method? Perhaps using PCA components instead of all images or doing
some preliminary feature elimination prior to starting with the
training. In the past years, there has been significant work towards
parsimony in supervised machine learning classifiers such as RF which
have been shown to produce classifying models with maximal accuracy
with only a few but discriminate hand-crafted features, almost
eliminating training time and decreasing workloads [R3]. In the
authors case, it seems that a SoA machine learning implementation
(ie., XgBoost instead of RF) with parallelization and a good feature
engineering could produce similar results with a minimal computational
effort than the range of CNN’s tested here. I could stress that future
efforts will perhaps combine the 2 approaches – a supervised ML to
produce fast and acceptable dense predictions and an FCN to train on
these predictions to further improve and to justify its use.
5.1 Answer
──────────
A discussion section has been added in order to take into accound this
remark.
Reviewer 2 Report
The manuscript proposes an adaptation of Convolutional Neural Networks and evaluates its performance for large scale land cover classification. Although the topic is interesting for the journal Remote Sensing, the presentation of the work is poor. The manuscript requires a full revision. Especial attention should be given to the introduction to properly state the problem and research question. The manuscript also has a very limited discussion of results and incomplete methods and data description.
Abstract - Please revise the abstract. The sentences are disconnect and do not introduce properly the problem and research question.
Section 1.1 – The first paragraph misses a closing statement.
Line 102 – replace “an class” with “a class”
Section 1.5 – In the first paragraph are you talking about HPC or reference data? Please revise the text.
Introduction – The introduction must be completely revised. The subsections of the introduction are not coherent and it is very hard to get what the authors want to tell.
Line 149 - “will be used” or “were used”? Usually, methods are written in the past tense.
Line 160 – The reader is not supposed to read a second paper to understand which data or methods you have used. Please, add the full description of your methods and data to the manuscript.
Line 174 – Is the accuracy evaluated per polygon or pixel? How many sample/pixels for each class were used for training and validation?
Line 197 – What is FG-UNET? Please, define the abbreviations before using them.
Section 3.1 – This section introduces lots of variables and equations which are not well explained in the manuscript. The reader has a really hard time to understand them.
Line 213 – What are the main differences from FG-UNET to UNET? It would be easier for the reader if you can explain them in the manuscript.
Line 279 – How many nodes and CPUs did you use? Please add to the manuscript.
Table 2 is very interesting, how much time would be required to train a model to the whole of Europe? This could be added to the discussion.
Author Response
━━━━━━━━━━━━━━━━━━━━━━━━━
RESPONSES TO REVIEWER 2
━━━━━━━━━━━━━━━━━━━━━━━━━
1 Comments and Suggestions for Authors
══════════════════════════════════════
The manuscript proposes an adaptation of Convolutional Neural Networks
and evaluates its performance for large scale land cover
classification. Although the topic is interesting for the journal
Remote Sensing, the presentation of the work is poor. The manuscript
requires a full revision. Especial attention should be given to the
introduction to properly state the problem and research question. The
manuscript also has a very limited discussion of results and
incomplete methods and data description.
1.1 Answer
──────────
The remarks of 3 reviewers have been taken into account in order to
improve the article. A point by point response is given below. We
thank the anonymous reviewer for the pertinent comments.
2 Abstract Please revise the abstract.
═══════════════════════════════════════
The sentences are disconnect and do not introduce properly the problem
and research question.
2.1 Answer
──────────
The abstract has been rewritten in order to take into account this
issue.
3 Section 1.1 – The first paragraph misses a closing statement.
═══════════════════════════════════════════════════════════════
3.1 Answer
──────────
We don't see the issue in the paragraph.
4 Line 102 – replace “an class” with “a class”
══════════════════════════════════════════════
4.1 Answer
──────────
Done
5 Section 1.5 –
═══════════════
In the first paragraph are you talking about HPC or reference data?
Please revise the text.
5.1 Answer
──────────
The text has been modified to make this clear.
6 Introduction
══════════════
The introduction must be completely revised. The subsections of the
introduction are not coherent and it is very hard to get what the
authors want to tell.
6.1 Answer
──────────
Important changes to the introduction (also taking into account other
reviewers' comments) have been made.
7 Line 149 -
════════════
“will be used” or “were used”? Usually, methods are written in the
past tense.
7.1 Answer
──────────
Corrected
8 Line 160 –
════════════
The reader is not supposed to read a second paper to understand which
data or methods you have used. Please, add the full description of
your methods and data to the manuscript.
8.1 Answer
──────────
We understand this remark. However, we chose not to do that because in
a recent submission of another connected work which used the same data
and the same pre-processing, publication was denied because of the
description of the data set:
This article has reused significant portion of writings in
an article published in MDPI Remote Sensing. This is
inappropriate.
In light of the appropriateness of your manuscript for our
journal, your manuscript has been denied publication.
Since the cited article is Open Access, we think that reading the
appropriate sections would be equivalent to including them here.
9 Line 174 –
════════════
Is the accuracy evaluated per polygon or pixel? How many sample/pixels
for each class were used for training and validation?
9.1 Answer
──────────
The validation is performed at the pixel level. As stated at the end
of the paragraph, 67% was used for training and 33% for validation.
The number of pixels (in hectares, 1 ha = 100 pixels) is given in
table 1. The text has been revised to make this clear.
10 Line 197 –
═════════════
What is FG-UNET? Please, define the abbreviations before using them.
10.1 Answer
───────────
Done
11 Section 3.1 –
════════════════
This section introduces lots of variables and equations which are not
well explained in the manuscript. The reader has a really hard time to
understand them.
11.1 Answer
───────────
The section has been rewritten to make things more clear.
12 Line 213 –
═════════════
What are the main differences from FG-UNET to UNET? It would be easier
for the reader if you can explain them in the manuscript.
12.1 Answer
───────────
The introduction of this section has been rewritten in order to give
an overview of the differences before proceeding to their detailed
description.
13 Line 279 –
═════════════
How many nodes and CPUs did you use? Please add to the manuscript.
13.1 Answer
───────────
Done.
14 Table 2
══════════
Is very interesting, how much time would be required to train a model
to the whole of Europe? This could be added to the discussion.
14.1 Answer
───────────
The EU has 4.5 million km^2. Our tiles have 10000 km^2. We need 450
tiles, which is about 40 times what we have used here. We have added
this to the discussion section.
Reviewer 3 Report
This paper contains some relevant and novel work on land cover mapping using high spatial resolution time series of Sentinel-2 images. The methods used compare standard pixel-based machine learning with context-based neural networks, for a number of tiles in different regions of France. The paper is well written and contains some interesting results that are definitely worthy of publication, however more discussion of the results and their implications for national land cover mapping are needed.
Abstract
This needs to contain more information about the key results, conclusions and implications when detailing the contributions of the work
Introduction
line 45 - give examples of how and why the previous work is unsatisfactory
line 63 - expand on the important confusions
line 65 - can classes be distinguished if they are similar, does this refer to spectral similarity?
line 87 - what did these authors conclude?
lines 88-97 - some clarification and citations needed
line 104 - how successful was this?
lines 105-118 - again more needed on the conclusions from these papers e.g. the level of improvement in accuracy etc
Data
line 144 - how many images per tile, and in later discussion need to comment on how the number and spread through the year might have implications for the results (e.g. mountain scenes have fewer images due to cloud cover, some phenological stages may be missed thus resulting in confusion between vegetation classes)
line 160 - while the detail of the reference set does not need to be replicated from [8] a couple of additional sentences on the source and date of the data would be helpful
Figure 1 - a legend is needed for the different colour regions, the tile labels also lack the T that all the references to tiles in the text include - need to be consistent
Table 1 - a final column indicating the percentage of each land cover type across all the tiles might also be useful, a one sentence summary of each tile might also be useful for those readers not familiar with the French landscape
Figure 2 - the text is very small and the numbers virtually impossible to read
Experimental setup
line 410 - these results, and others such as in line 424 need to be discussed in the results section
Results
line 448 - why is RF higher for T30TYN, is it because of the classes in that tile, or the number of images... conversely why is T31UDQ so much better for FG-UNET? These differences need to be teased out as there would be implications if this was applied at a national scale
line 449 - explain why tile T30TWT was chosen for the more detailed analysis - this only covers two eco-climatic zones and therefore has limited representativeness of the whole country, T31TDJ however covers six
line 450 - some discussion of why NGL is better mapped using RF needed, especially given the proportion of NGL in the dataset (sixth largest)
line 457 - more discussion needed on why the results vary between the five different runs, and what makes some classes more variable than others e.g. why is CUF so different? As commented for line 448 there could be implications for national scale mapping where only one run might be viable due to data processing constraints
line 458 - why is this - discuss as well as present results
Figure 4 - hard to justify this as a figure in its own right given that it is just the legend for figures 7 and 10
Figure 5 - can all the Kappa comparison plots take a y-axis from 0.5-1 in order to highlight the detail? and for all the Kappa and Fscore plots the text needs to be bigger
line 466 - in the same way that roads are treated in detail, so could other absent but important) classes be considered e.g. forests are defined as either broad-leaved or coniferous but are there some that could be mixed?
Figure 7 - a more detailed caption would help the reader, especially given the legend is on a different page
line 476 - discuss why the mountain tiles in particular are so much better
Figures 5-15 (minus 7 and 10) contain a lot of comparative information but it is hard to appreciate that from these graphs - is there an alternative way of highlighting some of the key differences?
line 516 - this section is very short and needs more discussion - in this instance the figure caption provides a lot more detail than the text, is the y-axis correct that only up to 0.5% of corners are correctly matched, should it be 50%? Given that potentially 90% or more of the corners (and not just "some" as defined in the text) are not matched by the FG-Unet approach the limitations of this are important - how much of this discrepancy might be reflected in the Kappa/Fscore results?
Conclusion
This section needs to be expanded with more insight into some of the implications of the results and what they mean for national land cover mapping programmes, e.g. the quality of training data is mentioned, but presumably this would need to be collected on an annual basis if this was to become an operational process, or could an FG-Unet approach learn from a baseline year?
Throughout there are some typos, data are not always referred to in the plural form, there is some inconsistency in tense e.g. in 2.1. there is past, present and future tense all in one paragraph, likewise there are some inconsistencies in the formatting of the references.
Author Response
━━━━━━━━━━━━━━━━━━━━━━━━━
RESPONSES TO REVIEWER 3
━━━━━━━━━━━━━━━━━━━━━━━━━
1 Comments and Suggestions for Authors
══════════════════════════════════════
This paper contains some relevant and novel work on land cover mapping
using high spatial resolution time series of Sentinel-2 images. The
methods used compare standard pixel-based machine learning with
context-based neural networks, for a number of tiles in different
regions of France. The paper is well written and contains some
interesting results that are definitely worthy of publication, however
more discussion of the results and their implications for national
land cover mapping are needed.
1.1 Answer
──────────
We thank the reviewer for the detailed comments and remarks. All of
them are taken into account in the point by point response below.
2 Abstract
══════════
This needs to contain more information about the key results,
conclusions and implications when detailing the contributions of the
work
2.1 Answer
──────────
The abstract has been rewrittent to take into account this remark.
3 Introduction
══════════════
• ☑ line 45 - give examples of how and why the previous work is
unsatisfactory
• done
• ☑ line 63 - expand on the important confusions
• done
• ☑ line 65 - can classes be distinguished if they are similar, does
this refer to spectral similarity?
• precision added
• ☑ line 87 - what did these authors conclude?
• conclusions added
• ☑ lines 88-97 - some clarification and citations needed
• a figure (figure 2 in the new version) has been added to
illustrate the problem of sparse annotations
• ☑ line 104 - how successful was this?
• conclusions added
• ☑ lines 105-118 - again more needed on the conclusions from these
papers e.g. the level of improvement in accuracy etc
• complementary information added
4 Data
══════
4.1 line 144 -
──────────────
how many images per tile, and in later discussion need to comment on
how the number and spread through the year might have implications for
the results (e.g. mountain scenes have fewer images due to cloud
cover, some phenological stages may be missed thus resulting in
confusion between vegetation classes)
4.1.1 Answer
╌╌╌╌╌╌╌╌╌╌╌╌
The number of available images has been added to the text. However, we
do not develop the analysis of errors as a function of image
availability, since the aim of the paper is comparing methods using
the same data set.
4.2 line 160 -
──────────────
while the detail of the reference set does not need to be replicated
from [8] a couple of additional sentences on the source and date of
the data would be helpful
4.2.1 Answer
╌╌╌╌╌╌╌╌╌╌╌╌
More details on the origin of the reference data have been added.
4.3 Figure 1 -
──────────────
a legend is needed for the different colour regions, the tile labels
also lack the T that all the references to tiles in the text include -
need to be consistent
4.3.1 Answer
╌╌╌╌╌╌╌╌╌╌╌╌
Done
4.4 Table 1 -
─────────────
a final column indicating the percentage of each land cover type
across all the tiles might also be useful, a one sentence summary of
each tile might also be useful for those readers not familiar with the
French landscape
4.4.1 Answer
╌╌╌╌╌╌╌╌╌╌╌╌
Done
4.5 Figure 2 -
──────────────
the text is very small and the numbers virtually impossible to read
4.5.1 Answer
╌╌╌╌╌╌╌╌╌╌╌╌
The size of the figure has been increased
5 Experimental setup
════════════════════
line 410 - these results, and others such as in line 424 need to be
discussed in the results section
5.1 Answer
──────────
Done
6 Results
═════════
6.1 line 448 - why is RF higher for T30TYN,
───────────────────────────────────────────
is it because of the classes in that tile, or the number of images…
conversely why is T31UDQ so much better for FG-UNET? These differences
need to be teased out as there would be implications if this was
applied at a national scale
6.1.1 Answer
╌╌╌╌╌╌╌╌╌╌╌╌
These differences are now explained in the text.
6.2 line 449 -
──────────────
Explain why tile T30TWT was chosen for the more detailed analysis -
this only covers two eco-climatic zones and therefore has limited
representativeness of the whole country, T31TDJ however covers six
6.2.1 Answer
╌╌╌╌╌╌╌╌╌╌╌╌
This tile was chosen because of the better balance between classes,
allowing a more detailed analysis.
6.3 line 450 -
──────────────
Some discussion of why NGL is better mapped using RF needed,
especially given the proportion of NGL in the dataset (sixth largest)
6.3.1 Answer
╌╌╌╌╌╌╌╌╌╌╌╌
The explanation is that for this class, context and geometry is less
important than spectral and temporal information.
6.4 line 457 -
──────────────
More discussion needed on why the results vary between the five
different runs, and what makes some classes more variable than others
e.g. why is CUF so different? As commented for line 448 there could be
implications for national scale mapping where only one run might be
viable due to data processing constraints
6.4.1 Answer
╌╌╌╌╌╌╌╌╌╌╌╌
The variability is higher for minority classes. One may expect that
this effect diminishes when working at the national scale. Some
discussion on this phenomena is added.
6.5 line 458 -
──────────────
Why is this - discuss as well as present results
6.5.1 Answer
╌╌╌╌╌╌╌╌╌╌╌╌
Urban areas are textured and context is more important than for
vegetation classes, where the spectral and temporal signatures convey
more information. This is added to the text.
6.6 Figure 4 -
──────────────
Hard to justify this as a figure in its own right given that it is
just the legend for figures 7 and 10
6.6.1 Answer
╌╌╌╌╌╌╌╌╌╌╌╌
We understand the remark, but since the legend is rather large, we
made the choice of removing it from the images so that they can be
larger and the details can be observed.
6.7 line 466 -
──────────────
In the same way that roads are treated in detail, so could other
absent but important) classes be considered e.g. forests are defined
as either broad-leaved or coniferous but are there some that could be
mixed?
6.7.1 Answer
╌╌╌╌╌╌╌╌╌╌╌╌
The reviewer highlights an interesting question. At 10 m resolution,
most pixels are pure and pixel-based classifiers would not be able to
identify mixed forests, while contextual classifiers as CNN might be
able to. However, the degree of mixture is also an issue. For some
land cover nomenclatures, mixed forests are only those with a high
degree of mixture (i.e. every other tree could be of a different
type), while for others, a forest stand made of large patches of
homogeneous types of trees are considered mixed. This is a
nomenclature issue which is beyond of the scope of this paper, for
which we use a pre-defined nomenclature.
6.8 Figure 7 -
──────────────
A more detailed caption would help the reader, especially given the
legend is on a different page
6.8.1 Answer
╌╌╌╌╌╌╌╌╌╌╌╌
Done
6.9 line 476 -
──────────────
Discuss why the mountain tiles in particular are so much better
6.9.1 Answer
╌╌╌╌╌╌╌╌╌╌╌╌
This is the same phenomenon observed when comparing FG-UNET and RF:
pixel-based classification works better for this areas and FG-UNET
adds a pixel-based path which complements the contextual
classification. This remark is added in the text.
6.10 Figures 5-15 (minus 7 and 10)
──────────────────────────────────
Contain a lot of comparative information but it is hard to appreciate
that from these graphs - is there an alternative way of highlighting
some of the key differences?
6.10.1 Answer
╌╌╌╌╌╌╌╌╌╌╌╌╌
Another reviewer suggested a tabular representation of all the
results, but this was less readable that the graphical form. The key
differences are highlighted in the text.
6.11 line 516 -
───────────────
This section is very short and needs more discussion - in this
instance the figure caption provides a lot more detail than the text,
is the y-axis correct that only up to 0.5% of corners are correctly
matched, should it be 50%? Given that potentially 90% or more of the
corners (and not just "some" as defined in the text) are not matched
by the FG-Unet approach the limitations of this are important - how
much of this discrepancy might be reflected in the Kappa/Fscore
results?
6.11.1 Answer
╌╌╌╌╌╌╌╌╌╌╌╌╌
The figure has been correctly labelled with "proportion" instead of
"percentage". The text has also been rewritten in order to note that
the pixel based classifications also match less than half of the
corners.
As explained in the description of the measure (Experimental Setup
section) this discrepancy is not observed in the standard metrics,
since the boundary and corner pixels represent a very low proportion
of the validation pixels which reside mainly inside the objects.
7 Conclusion
════════════
7.1 This section needs to be expanded
─────────────────────────────────────
with more insight into some of the implications of the results and
what they mean for national land cover mapping programmes, e.g. the
quality of training data is mentioned, but presumably this would need
to be collected on an annual basis if this was to become an
operational process, or could an FG-Unet approach learn from a
baseline year?
7.1.1 Answer
╌╌╌╌╌╌╌╌╌╌╌╌
A discussion section has been added before the conclusions.
7.2 Throughout there are some typos,
────────────────────────────────────
data are not always referred to in the plural form, there is some
inconsistency in tense e.g. in 2.1. there is past, present and future
tense all in one paragraph, likewise there are some inconsistencies in
the formatting of the references.
7.2.1 Answer
╌╌╌╌╌╌╌╌╌╌╌╌
These issues have been corrected.